# 🤖 CogVLA: Cognition-Aligned Vision-Language-Action Model via Instruction-Driven Routing & Sparsification

**Wei Li    Renshan Zhang    Rui Shao**[*]    **Jie He    Liqiang Nie**

School of Computer Science and Technology, Harbin Institute of Technology, Shenzhen

liwei2024@stu.hit.edu.cn     shaorui@hit.edu.cn

https://jiutian-vl.github.io/CogVLA-page

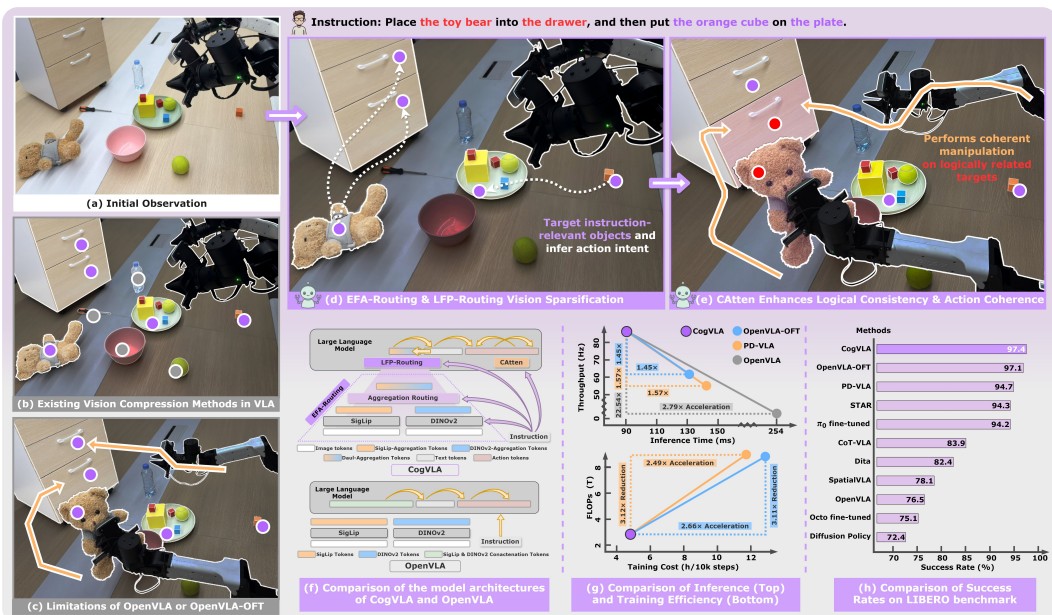

Figure 1: **Overview of our proposed CogVLA.** Traditional VLA models process initial observations (Fig.**(a)**) without vision compression, leading to high computational cost. As shown in Fig.**(b)** and Fig.**(d)**, existing compression methods retain irrelevant inputs ⚫ and fail to focus on instruction-relevant targets 🟣. CogVLA employs EFA-Routing and LFP-Routing to sparsify visual inputs based on instruction relevance. Comparing Fig.**(c)** and Fig.**(e)**, CAtten further enhances logical consistency and action coherence for final targeted objects 🔴. Fig.**(f)**, Fig.**(g)**, and Fig.**(h)** illustrate the architectural innovations of CogVLA and its superiority in efficiency and performance.

## Abstract

Recent Vision-Language-Action (VLA) models built on pre-trained Vision-Language Models (VLMs) require extensive post-training, resulting in high computational overhead that limits scalability and deployment. Existing sparsification strategies—such as Mixture-of-Depths, layer skipping, and early exit—fall short by neglecting the semantic coupling across vision-language-action modalities,

---

[*]Corresponding author

39th Conference on Neural Information Processing Systems (NeurIPS 2025).

and focusing narrowly on intra-LLM computation while overlooking end-to-end coherence from perception to control. To address these challenges, we propose **CogVLA**, a Cognition-Aligned Vision-Language-Action framework that leverages instruction-driven routing and sparsification to improve both efficiency and performance. CogVLA draws inspiration from human multimodal coordination and introduces a 3-stage progressive architecture. 1) **Encoder-FiLM based Aggregation Routing (EFA-Routing)** injects instruction information into the vision encoder to selectively aggregate and compress dual-stream visual tokens, forming a instruction-aware latent representation. 2) Building upon this compact visual encoding, **LLM-FiLM based Pruning Routing (LFP-Routing)** introduces action intent into the language model by pruning instruction-irrelevant visually grounded tokens, thereby achieving token-level sparsity. 3) To ensure that compressed perception inputs can still support accurate and coherent action generation, we introduce **V-L-A Coupled Attention (CAtten)**, which combines causal vision-language attention with bidirectional action parallel decoding. Extensive experiments on the LIBERO benchmark and real-world robotic tasks demonstrate that CogVLA achieves state-of-the-art performance with success rates of 97.4% and 70.0%, respectively, while reducing training costs by $2.5\times$ and decreasing inference latency by $2.8\times$ compared to OpenVLA.

# 1 Introduction

Vision-Language Action (VLA)[88, 36, 30, 6, 84, 85] research has advanced rapidly, fueled by the rich visual and linguistic representations provided by powerful pre-trained Vision-Language Models (VLMs)[35, 1, 42, 22, 2, 78]. Leveraging these foundational models, the VLA paradigm is progressing toward end-to-end robotic control and embodied intelligence, enabling agents to comprehend natural language instructions, perceive complex scenes, and perform manipulation tasks with minimal task-specific engineering. Pioneering works such as RT-2 [8], Octo [65], OpenVLA [30], $\pi_0$ [6], and $\pi_{0.5}$ [27] have demonstrated the potential of this paradigm.

However, aligning the high-dimensional multimodal features output by VLMs with continuous action spaces remains **computationally expensive** [87, 71, 10, 5, 30]. Standard fine-tuning and joint training procedures often entail substantial memory consumption, high FLOPs overhead, and extended training times, severely limiting scalability and practical deployment on resource-constrained platforms. For instance, fine-tuning a 7B VLA model [29] with action chunking on a single-task dataset from the LIBERO benchmark [41] consumes over 600 GPU hours (using 80G A100 GPUs), incurring significant computational costs. Although techniques such as Mixture-of-Depths [51, 77, 46, 13], layer skipping [82, 72], and early exit [74, 17] have been proposed to sparsify and accelerate the model training and inference, these methods primarily focus on computation optimization within language models, overlooking the semantic coupling across perception, language alignment, and action decoding. This modular optimization paradigm often leads to **cross-modal semantic degradation**, manifesting as follows: **i)** visual compression within encoders discards task-relevant fine-grained features, **ii)** token skipping within LLMs disrupts the contextual coherence necessary for reference resolution, and **iii)** action generation lacks causal reasoning over multimodal state transitions.

From a cognitive science perspective [52, 32], humans exhibit a highly optimized and efficient multimodal coordination mechanism during manipulation. For example, when receiving the instruction "place the red cup at the corner of the table," the human *Visual and Attention System (VAS)* selectively focuses [7] on the color attributes of the cup and the spatial structure of the table. Concurrently, the *Supplementary Motor Area (SMA)* injects task-relevant action intentions [61] derived from key semantic associations (e.g., "red–cup–corner") into the visual processing stream, while the *Premotor Cortex (PMC)* dynamically integrates both visual and linguistic information to plan coherent motion trajectories. This organic unification of perception, reasoning, and control results in remarkable task efficiency. Inspired by this, we propose **CogVLA**, a Cognition-Aligned Vision-Language-Action framework based on Instruction-Driven Routing & Sparsification, as shown in Fig. 1 (f). Unlike existing modular pipelines, CogVLA establishes a task-semantic-consistent joint optimization mechanism across vision, language, and action modalities, reinforcing cross-modal coherence while improving computational efficiency.

Specifically, CogVLA adopts a 3-stage progressive design to jointly enhance computational efficiency and task performance, as shown in Fig. 1 (d) and (e): **1) Encoder-FiLM based Aggregation Routing (EFA-Routing)**: To alleviate visual information redundancy and achieve VAS-like visual focus, EFA-Routing compresses visual tokens to 25% of the original input scale, guided by task-specific instructions. This process begins by dynamically encoding the instruction into modulation parameters that guide the aggregation of visual tokens within the visual encoder. Subsequently, the outputs from different encoder branches are adaptively fused to produce cross-branch representations that are semantically aligned with the given task. **2) LLM-FiLM based Pruning Routing (LFP-Routing)**: Building upon the aggregated visual encoding, LFP-Routing learns a novel, instruction-aware sparsity pattern to prune visual tokens within the language model. By emulating the functionality of SMA, which injects action intentions into visual features, the mechanism selectively skips attention computations over 50% of task-irrelevant tokens. As a result, it significantly reduces the computational burden of the language model and effectively minimizes latency in action generation. **3) V-L-A Coupled Attention (CAtten):** To ensure that the compressed visual inputs retain the capacity to support accurate and coherent action sequence, CAtten introduces a coupled attention mechanism inspired by PMC: **i)** Cross-modal causal attention is applied between the V-L-A interaction layer to preserve temporal reasoning capabilities; **ii)** Unidirectional attention is employed within the V-L layer to ensure semantic consistency, where visual features have been pre-enhanced with task-specific language intent; **iii)** Bidirectional attention is utilized within the Action layer to enhance the coherence of action sequences and enable efficient parallel decoding.

We conduct comprehensive evaluations of CogVLA on the LIBERO benchmark and real-world robotic manipulation tasks. Experimental results show that CogVLA achieves state-of-the-art task success rates while reducing end-to-end computational costs significantly, as shown in Fig. 1 (g) and (h). Ablation studies further validate the complementarity and synergistic effect of the routing modules and the coupled attention mechanism. Our main contributions are summarized as follows:

- We propose **CogVLA**, a Cognition-Aligned Vision-Language-Action framework inspired by human multimodal coordination, which establishes a biomimetic 3-stage architecture: "*VAS (visual information focusing) – SMA (semantic intent filtering) – PMC (action sequence planning)*."

- We develop synergistic **EFA-Routing** and **LFP-Routing**, enabling instruction-driven vision sparsification in perception-reasoning pipelines.

- We formulate **CAtten** ensuring cross-modal logical consistency and temporal action coherence in doubly compressed multimodal representations.

- Through extensive experiments on the LIBERO benchmark and real-world robotic tasks, we demonstrate the superior performance and efficiency of CogVLA.

## 2 Methods

### 2.1 Preliminary: Parallel Decoding in Action Chunk

We consider a sequence prediction setting where a Vision-Language-Action (VLA) model outputs a sequence of actions across $K$ future timesteps. Traditional *autoregressive (AR) decoding* predicts actions sequentially, whereas *parallel decoding* enables simultaneous prediction of all actions within the chunk, improving inference efficiency and supporting scalable deployment.

**Action Chunk.** Given the current input context $\mathbf{X} = \{\mathbf{I}, \mathbf{t}\} \in \mathbb{R}^{M+T}$, which comprises the visual observation and task instruction, the model predicts a chunk of $K$ future actions:

$$\mathbf{A} = [\mathbf{a}_0, \mathbf{a}_1, \ldots, \mathbf{a}_{K-1}] \in \mathbb{R}^{K \times D} \tag{1}$$

Here, $D$ denotes the dimensionality of each atomic action (e.g., $D = 7$ for 3-DoF translation $\Delta T$, 3-DoF rotation $\Delta R$, and binary gripper control).

**Autoregressive Decoding.** In causal autoregressive decoding, the action sequence is generated incrementally. For each timestep $i \in 0, \ldots, K-1$, the atomic action vector $\mathbf{a}_i \in \mathbb{R}^D$ is produced token-by-token, with each token $\mathbf{a}_i^{(k)}$ conditioned on the preceding tokens and previously actions:

$$\mathbf{a}_i = [\mathbf{a}_i^{(1)}, \mathbf{a}_i^{(2)}, \cdots, \mathbf{a}_i^{(D)}]^\top, \quad \mathbf{a}_i^{(k)} = f_{\text{AR}}([\mathbf{X}, \{\mathbf{a}_\tau\}_{\tau < i}, \mathbf{a}_i^{(1:k-1)}]) \tag{2}$$

This decoding necessitates $K \times D$ forward passes, introducing latency from token-level dependencies.

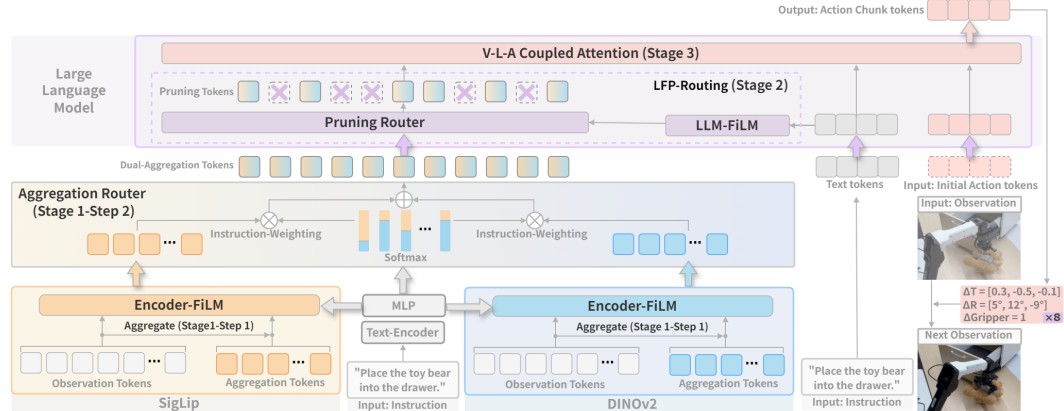

Figure 2: **Overview of CogVLA Framework.** CogVLA employs a cognition-aligned, instruction-driven routing & sparsification strategy for efficient action chunk prediction. Inspired by human multimodal coordination, it integrates task-guided visual aggregation, semantic pruning, and coherent decoding, ensuring efficient cross-modal representation alignment from perception to control.

**Parallel Decoding.** In contrast, parallel decoding eliminates the sequential dependency. The model receives the input observation embeddings $\mathbf{X}$ along with $K$ empty placeholder embeddings:

$$\tilde{\mathbf{X}} = [\mathbf{X}, \mathbf{0}_0, \mathbf{0}_1, \ldots, \mathbf{0}_{K-1}] \in \mathbb{R}^{M+T+K \times D} \tag{3}$$

where $\mathbf{0}_i \in \mathbb{R}^D$ denotes a learnable zero-action embedding. Under a *bidirectional attention* scheme (instead of causal masking), the decoder jointly produces all future actions in a single pass:

$$\mathbf{A} = f_{\text{parallel}}(\tilde{\mathbf{X}}) \tag{4}$$

## 2.2 CogVLA: Framework

Recent methods [74, 82, 49, 62, 29] primarily focus on lightweight computation within isolated stages of action chunk prediction in VLA models, often leading to cross-modal semantic degradation due to modular disconnection. To address this, we propose **CogVLA**, a cognition-aligned framework that enhances both efficiency and performance via Instruction-Driven Routing & Sparsification.

As illustrated in Fig. 2, the framework operates through a 3-stage progressive architecture inspired by human multimodal coordination. In **Stage 1**, CogVLA incorporates $N$ vision encoders $\{\text{Enc}_1, \ldots, \text{Enc}_N\}$ that extract visual tokens from image observations $\mathbf{I}^{(i)}$. Each encoder is modulated by the instruction $\mathbf{t}_r$ (obtained via LLM embedding layer) through an Encoder-FiLM module:

$$\mathbf{v}_{\text{agg}}^{(i)} = \text{Encoder-FiLM}_i(\mathbf{I}^{(i)}, \mathbf{v}_{\text{agg}}^{(i)}, \mathbf{t}_r), \quad i = 1, ..., N \tag{5}$$

where $\mathbf{v}_{\text{agg}}^{(i)}$ denotes the aggregation token for the $i$-th encoder. These modality-specific aggregated tokens are dynamically dual-aggregated via an instruction-conditioned routing mechanism:

$$\mathbf{v}_{\text{agg}} = \sum_{i=1}^{M} \alpha_i \cdot \mathbf{v}_{\text{agg}}^{(i)} \tag{6}$$

$$\boldsymbol{\alpha} = [\alpha_1, \ldots, \alpha_N]^\top = \text{Softmax}\left(\text{MLP}_{\text{route}}(\mathbf{t}_r)\right) \tag{7}$$

In **Stage 2**, the dual-aggregation tokens are injected into the LLM, where LFP-Routing selectively filters out instruction-irrelevant visual tokens. This instruction-driven sparsification enables token-level efficiency by reducing redundant attention computation, aligning the retained tokens with task-relevant semantics. The filtered representation is then processed by the proposed CAtten module to produce the task-aligned representation in **Stage 3**. The module operates as follows:

$$\mathbf{Z}_{l+1} = \text{CAtten}(\text{LFP-Routing}(\mathbf{Z}_l, \mathbf{t}_l)) \tag{8}$$

where $\mathbf{Z}_l$, and $\mathbf{t}_l$ denote the visual and instruction input tokens at the $(l+1)$-th transformer layer, respectively, with $\mathbf{Z}_0 = \mathbf{v}_{\text{agg}}$ as the initial state. Finally, action chunks are decoded in parallel using the compressed multimodal context combined with placeholder action tokens:

$$\mathbf{A}_t = f_{\text{parallel}}(\tilde{\mathbf{X}}) = f_{\text{parallel}}([\mathbf{Z}_0, \mathbf{t}_0, \mathbf{0}_0, \mathbf{0}_1, \ldots, \mathbf{0}_{K-1}]) \tag{9}$$

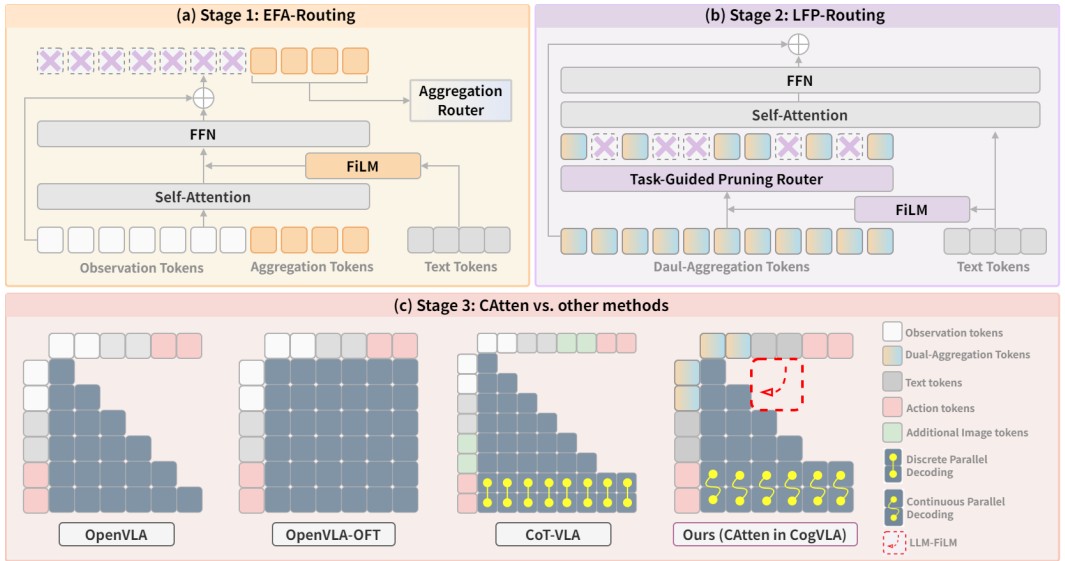

Figure 3: **Illustration of 3-Stage Progressive Design.** CogVLA emulates human multimodal coordination via instruction-driven routing and sparsification. EFA-Routing (Stage 1), LFP-Routing (Stage 2), and CAtten (Stage 3) correspond to the VAS, SMA, and PMC, respectively. Fig.(c) highlights the advantages of CAtten over prior attention mechanisms in combining uni-&bi-directional attention, injecting action intent, enabling parallel decoding, and leveraging sparse visual tokens.

Through this progressive design across **Stage 1–3**, CogVLA realizes instruction-driven sparsification and routing across the vision-language-action pipeline, effectively reducing computational overhead while preserving task-relevant semantics and enhancing cross-modal reasoning fidelity.

## 2.3 CogVLA: Cognition-Aligned 3-Stage Progressive Design

As illustrated in Fig. 3, CogVLA adopts a **3-Stage Progressive Design**, which emulates the human's optimized coordination during manipulation tasks: **EFA-Routing** mimics the *VAS* by selectively aggregating visual tokens conditioned on task-specific instructions, thereby achieving focused perception. **LFP-Routing** emulates the *SMA* by introducing action intentions into the visual context within the language model, selectively pruning irrelevant tokens to achieve instruction-driven token sparsity. **CAtten** simulates the *PMC* by dynamically integrating compressed multimodal representations, ensuring cross-modal logical consistency and temporal coherence in action decoding.

### 2.3.1 Encoder-FiLM based Aggregation Routing

**Step 1: Intra-encoder Aggregation.** To aggregate visual information and enable instruction-guided representation learning, we introduce Encoder-FiLM, which dynamically consolidates observation tokens into aggregation tokens based on task-specific instructions. The language instruction $\mathbf{t}_r$ modulates visual tokens $\mathbf{I}^{(i)}$ and aggregation tokens $\mathbf{v}_{\text{agg}}^{(i)}$ within each visual encoder branch:

$$f_{\text{FA}}(\mathbf{I}^{(i)}, \mathbf{v}_{\text{agg}}^{(i)}, \mathbf{t}_r) = (\mathbf{1} + \gamma_i(\mathbf{t}_r)) \odot \text{Self-Att}(\mathbf{I}^{(i)}, \mathbf{v}_{\text{agg}}^{(i)}) + \beta_i(\mathbf{t}_r)$$
$$\mathbf{v}_{\text{agg}}^{(i)} = \text{Aggregate}(\text{FFN}(f_{\text{FA}}(\cdot))) + \mathbf{v}_{\text{agg}}^{(i)}$$

(10)

where $\gamma_i$ and $\beta_i$ denote the FiLM-generated scale and shift vectors conditioned on $\mathbf{t}_r$, and $\odot$ represents element-wise multiplication. Through iterative visual encoder blocks, the aggregation token $\mathbf{v}_{\text{agg}}^{(i)}$ adaptively integrates instruction-relevant information from observation tokens while discarding redundant information. Consequently, only the final $\mathbf{v}_{\text{agg}}^{(i)}$ is retained while the image tokens $\mathbf{I}^{(i)}$ are discarded, effectively reducing the number of visual tokens to 25% of the original size.

**Step 2: Cross-encoder Aggregation.** To integrate the aggregated visual representations from two heterogeneous vision encoder branches (SigLIP and DINOv2), we design an instruction-conditioned aggregation routing gate that computes a fusion weight $\alpha \in (0, 1)$ based on the input language

instruction. Rather than statically assigning equal contributions, the fusion ratio is dynamically predicted for different tasks to reflect instruction-dependent visual preferences:

$$\alpha = \text{Sigmoid}(W_2(\sigma(W_1\mathbf{t}_r + \mathbf{b}_1)) + \mathbf{b}_2) \tag{11}$$

where $W_1$, $W_2$ are trainable weight matrices, $\mathbf{b}_1$, $\mathbf{b}_2$ are biases, and $\sigma$ denotes the GeLU [24] non-linearity. The final dual-aggregated visual token is computed as:

$$\mathbf{v}_{\text{agg}} = \alpha \cdot \mathbf{v}_{\text{agg}}^{\text{SigLIP}} + (1 - \alpha) \cdot \mathbf{v}_{\text{agg}}^{\text{DINOv2}} \tag{12}$$

This instruction-conditioned aggregation routing allows the model to adaptively balance visual features from different encoders based on the semantics of the instruction, promoting more effective cross-modal fusion, as shown in Fig. 2 and Fig. 3 (a). At the first transformer layer, we denote $\mathbf{Z}_0 = \mathbf{v}_{\text{agg}}$ as the dual-aggregation visual tokens and $\mathbf{t}_0$ as the instruction tokens, which serve as the initial inputs for **Stage 2**.

### 2.3.2 LLM-FiLM based Pruning Routing

Motivated by sparse token routing techniques [77, 46, 72], we recognize that EFA-Routing (Stage 1) aggregates features across all image tokens, potentially retaining redundant or semantically irrelevant visual information. To further reduce computational overhead and steer the visual representation toward the intended action semantics, we propose a lightweight LFP-Routing module prior to injecting visual context into the large language model, as illustrated in Fig. 3(b).

Given the dual-aggregation tokens and the corresponding task instruction at transformer layer $l$, denoted as $\mathbf{Z}_l$ and $\mathbf{t}_l$ respectively, LLM-FiLM performs a semantic-aware modulation as follows:

$$f_{\text{FP}}(\mathbf{Z}_l, \mathbf{t}_l) = \text{Prune}((\mathbf{1} + \gamma_{\text{LLM}}(\mathbf{t}_l)) \odot \mathbf{Z}_l) + \beta_{\text{LLM}}(\mathbf{t}_l))$$
$$\mathbf{Z}_{l+1} = \text{FFN}(\text{Self-Att}(f_{\text{FP}}(\cdot))) + \mathbf{Z}_l \tag{13}$$

where $\gamma_{\text{LLM}}(\cdot)$ and $\beta_{\text{LLM}}(\cdot)$ denote instruction-conditioned scaling and shifting functions, respectively, both implemented as lightweight MLPs. The Prune$(\cdot)$ operation selectively discards tokens with low task relevance, producing a filtered representation that maintains critical visual semantics.

We introduce a Task-Guided Pruning Router to implement the Prune$(\cdot)$ operation. This module filters tokens based on their instruction-aware relevance, preserving only those most critical to the task. At Transformer layer $l$, routing weights $R_l^j$ are computed for each visual token $\mathbf{Z}_l^j$ using an MLP:

$$R_l^j = \text{MLP}(\mathbf{Z}_l^j) \tag{14}$$

We define a token retention ratio $\beta$, and determine a relevance threshold $P_\beta^l$ as the $\beta$-th percentile of the routing weights at layer $l$. Only tokens whose scores exceed $P_\beta^l$ are preserved. Formally:

$$\mathbf{Z}_{l+1}^j = \begin{cases} R_l^j \times f_{\text{SF}}([\mathbf{Z}_l^j, \mathbf{t}_l]) + \mathbf{Z}_l^j, & \text{if } R_l^j > P_l^\beta \\ \mathbf{Z}_l^j, & \text{otherwise} \end{cases} \tag{15}$$

where $f_{\text{SF}}(\cdot)$ represents the self-attention and feed-forward operations within the current layer. The hyperparameter $\beta \in [0, 1]$ governs the sparsity level by controlling the proportion of retained tokens.

### 2.3.3 V-L-A Coupled Attention

To maintain semantic consistency and temporal coherence under compressed multimodal inputs, we introduce V-L-A Coupled Attention (CAtten), a biologically inspired mechanism grounded in the functional role of the *PMC* for planning and coordination. As shown in Fig. 3 (c), CAtten hierarchically combines causal and bidirectional attention across vision, language, and action streams. At the $l$-th transformer layer of the LLM, the input multimodal token sequence is defined as:

$$\tilde{\mathbf{X}} = [\mathbf{Z}_l, \mathbf{t}_l, \mathbf{A}_l] \in \mathbb{R}^{M+T+K \times D} \tag{16}$$

where $\mathbf{A}_l = [\mathbf{a}_0^l, \ldots, \mathbf{a}_{K-1}^l]$ denotes the action chunk. CAtten operates in three consecutive stages:

**Causal Vision-Language Attention.** To preserve instruction-conditioned visual reasoning, causal attention is applied over the concatenated vision-language token segment:

$$\text{Attn}_{\text{VL}}([\mathbf{Z}_l, \mathbf{t}_l]) = \text{Softmax}\left(\frac{[\mathbf{Z}_l, \mathbf{t}_l][\mathbf{Z}_l, \mathbf{t}_l]^\top}{\sqrt{d}} + \mathbf{M}_{\text{causal}}^{\text{VL}}\right)[\mathbf{Z}_l, \mathbf{t}_l] \tag{17}$$

where $\mathbf{M}_{\text{causal}}^{\text{VL}} \in \mathbb{R}^{(M+T) \times (M+T)}$ is a lower-triangular mask within vision-language tokens.

Table 1: **Simulation Experimental Results.** Comparison of task success rates (SR) and their ranks (RK) on the LIBERO benchmark across four task types. "†" indicates our reproduced results.

| Method | Spatial | | Object | | Goal | | Long | | Average | |
|---|---|---|---|---|---|---|---|---|---|---|
| | SR ↑ | RK ↓ | SR ↑ | RK ↓ | SR ↑ | RK ↓ | SR ↑ | RK ↓ | SR ↑ | RK ↓ |
| Diffusion Policy *[RSS'23]* [14] | 78.3 | 11 | 92.5 | 7 | 68.3 | 11 | 50.5 | 11 | 72.4 | 11 |
| Octo fine-tuned *[RSS'23]* [65] | 78.9 | 10 | 85.7 | 11 | 84.6 | 8 | 51.1 | 10 | 75.1 | 10 |
| OpenVLA *[CoRL'24]* [30] | 84.7 | 8 | 88.4 | 10 | 79.2 | 9 | 53.7 | 9 | 76.5 | 9 |
| $\pi_0$ fine-tuned *[RSS'25]* [6] | 96.8 | 3 | 98.8 | 1 | 95.8 | 3 | 85.2 | 5 | 94.2 | 5 |
| $\pi_0$-Fast *[RSS'25]* [49] | 96.4 | 5 | 96.8 | 6 | 88.6 | 7 | 60.2 | 7 | 85.5 | 7 |
| $\pi_{0.5}$-KI *[arXiv'25]* [18] | 98.0 | 2 | 97.8 | 5 | 95.6 | 4 | 85.8 | 4 | 96.0 | 4 |
| OpenVLA-OFT *[RSS'25]* [29] | 97.6 | 4 | 98.4 | 3 | **97.9** | **1** | 94.5 | 2 | 97.1 | 2 |
| SpatialVLA *[RSS'25]* [50] | 88.2 | 6 | 89.9 | 9 | 78.6 | 10 | 55.5 | 8 | 78.1 | 8 |
| PD-VLA† *[arXiv'25]* [62] | 95.5 | 6 | 96.7 | 7 | 94.9 | 6 | 91.7 | 3 | 94.7 | 3 |
| STAR *[ICML'25]* [23] | 95.5 | 7 | 98.3 | 4 | 95.0 | 5 | 88.5 | 6 | 94.3 | 6 |
| Dita *[arXiv'25]* [25] | 84.2 | 9 | 96.3 | 8 | 85.4 | 9 | 63.8 | 6 | 82.4 | 7 |
| CoT-VLA *[CVPR'25]* [84] | 87.5 | 7 | 91.6 | 8 | 87.6 | 8 | 69.0 | 6 | 83.9 | 6 |
| CogVLA | **98.6** | **1** | **98.8** | **1** | 96.6 | 2 | **95.4** | **1** | **97.4** | **1** |

Table 2: **Real-world Experimental Results.** Performance comparison on the Cobot Agilex ALOHA tasks. "†" indicates our reproduced results, while "*" denotes results reported in the original paper.

| Method | Object Placement | | Drawer Manipulation | | | T-shirt Folding | | | Average |
|---|---|---|---|---|---|---|---|---|---|
| | Cube→Plate | +Toy→Bowl | Open | +Place | +Close | Step 1 | +Step 2 | +Step 3 | SR |
| VQ-BeT [34]* | 5/10 | 3/10 | 4/10 | 3/10 | 1/10 | - | - | - | 20.0% |
| QueST [47]* | 6/10 | 4/10 | 3/10 | 1/10 | 0/10 | - | - | - | 20.0% |
| STAR* [23] | 8/10 | 6/10 | 6/10 | 4/10 | 3/10 | - | - | - | 45.0% |
| PD-VLA† [62] | 8/10 | 7/10 | 6/10 | 6/10 | 4/10 | 7/10 | 6/10 | 4/10 | 50.0% |
| OpenVLA-OFT† [29] | 8/10 | 7/10 | **8/10** | 6/10 | 5/10 | 7/10 | 7/10 | 5/10 | 56.7% |
| CogVLA | **9/10** | **8/10** | **8/10** | **7/10** | **7/10** | **9/10** | **8/10** | **6/10** | **70.0%** |

**Bidirectional Action Chunk Decoding.** To support coherent action generation, bidirectional attention is employed within the action decoding, allowing full context integration among future action tokens:

$$\text{Attn}_{\text{act}}(\mathbf{A}_l) = \text{Softmax}\left(\frac{\mathbf{A}_l \mathbf{A}_l^\top}{\sqrt{d}} + \mathbf{M}_{\text{bi}}^{\text{act}}\right) \mathbf{A}_l \qquad (18)$$

where $\mathbf{M}_{\text{bi}}^{\text{act}} \in \mathbb{R}^{(K \times D) \times (K \times D)}$ enables full continuous parallel decoding within each action chunk while maintaining causal dependencies from visual-language inputs.

**Unified Hybrid Attention Mask.** A global attention mask $\mathbf{M}_{\text{hybrid}} \in \mathbb{R}^{(M+T+K\times D) \times (M+T+K\times D)}$ enforces hierarchical token dependencies across vision, language, and action modalities:

$$\text{CAtten}(\tilde{\mathbf{X}}) = \text{Softmax}\left(\frac{\tilde{\mathbf{X}}\tilde{\mathbf{X}}^\top}{\sqrt{d}} + \mathbf{M}_{\text{hybrid}}\right) \tilde{\mathbf{X}} \qquad (19)$$

$$\mathbf{M}_{\text{hybrid}} = \begin{bmatrix} \mathbf{M}_{\text{causal}}^{\text{VL}} & -\infty & -\infty \\ \mathbf{0} & \mathbf{0} & -\infty \\ \mathbf{0} & \mathbf{0} & \mathbf{M}_{\text{bi}}^{\text{act}} \end{bmatrix} \qquad (20)$$

This coupled attention structure enables CogVLA to retain fine-grained cross-modal alignment and planning consistency under significant sparsification of visual inputs, ensuring that action sequences remain both semantically relevant and temporally coherent throughout the decoding process.

## 3 Experiments

### 3.1 Experiments Setting

All experiments are conducted on 4× A800 GPUs (80GB), benefiting from CogVLA's efficient instruction-driven sparsification. Implementation details are in **Appendix A**.

**Simulation Benchmark.** We use the LIBERO benchmark [41] to evaluate task performance and efficiency. Its long and diverse instructions (avg. 10.48 words vs. 3.34 in RLBench) reflect the

Table 3: **Efficiency Optimization Results.** CogVLA maintains superior performance while achieving the highest efficiency. Ablation studies on Stage 1 and Stage 2 validate the efficiency contribution of each routing module. "†" indicates our reproduced results.

| Method | Inference Time ↓ | Throughput ↑ | FLOPs ↓ | Taining Cost ↓ | LIBERO SR ↑ |
|---|---|---|---|---|---|
| OpenVLA† [30] | 0.254 s | 3.9 Hz | 8.48 T | 11.7 h/10k steps | 76.5% |
| OpenVLA-OFT† [29] | 0.132 s | 60.6 Hz | 8.45 T | 12.5 h/10k steps | 97.1% |
| PD-VLA† [62] | 0.143 s | 55.9 Hz | 8.48 T | 11.7 h/10k steps | 94.7% |
| CogVLA | **0.091 s** | **87.9 Hz** | **2.72 T** | **4.7 h/10k steps** | **97.4%** |
| *w/o* Stage 1 | 0.162 s | 49.4 Hz | 5.38 T | 8.4 h/10k steps | - |
| *w/o* Stage 2 | 0.117 s | 68.4 Hz | 3.52 T | 5.3 h/10k steps | - |

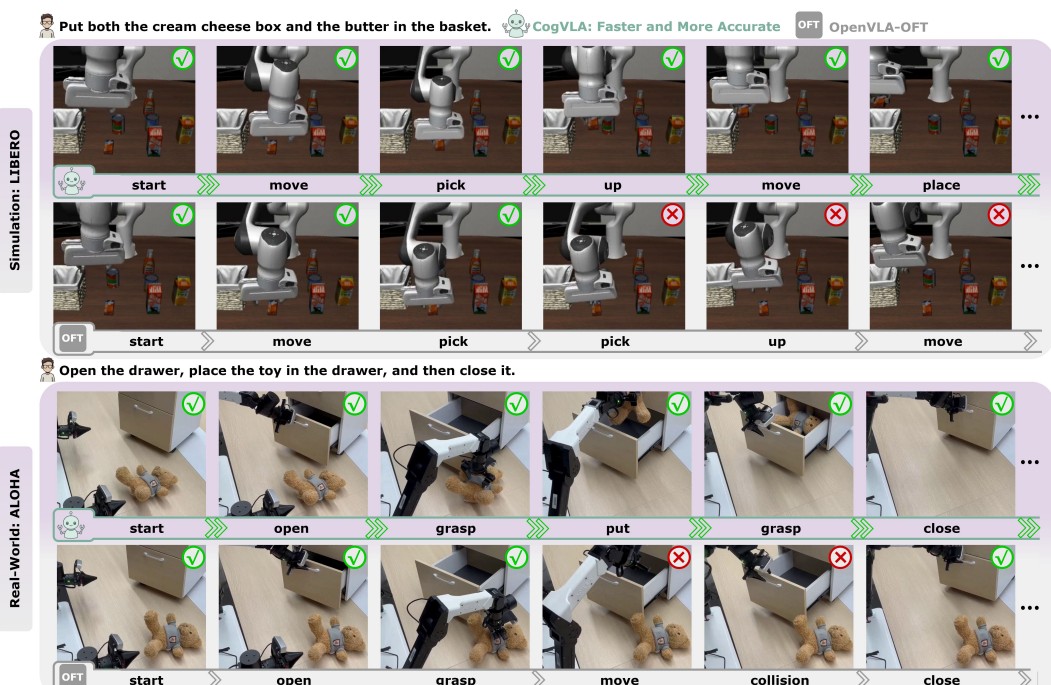

Figure 4: **Visualization comparison between CogVLA and OpenVLA-OFT.** CogVLA outperforms OpenVLA-OFT in success rates on both simulation and real-world tasks, achieving state-of-the-art performance with a 31% reduction in inference time. It also demonstrates superior training efficiency, requiring 3.1× fewer FLOPs and 2.7× shorter training time.

model's language understanding. LIBERO covers four suites—Spatial, Object, Goal, and Long—each with 10 tasks and 50 demonstrations.

**Real-World Experiments.** CogVLA is deployed on the Cobot Agilex ALOHA platform for three long-horizon tasks: Object Placement, Drawer Manipulation, and T-shirt Folding (45, 45, and 30 demonstrations). We introduce spatial and semantic variations during data collection.

**Baselines.** We compare CogVLA with multiple state-of-the-art methods, such as OpenVLA, SpatialVLA, STAR, and CoT-VLA. For efficiency assessment, we further evaluate OpenVLA, along with the top-performing PD-VLA and OpenVLA-OFT—an improved variant of OpenVLA that achieves higher performance and efficiency—under the same fine-tuning and inference settings as CogVLA.

## 3.2 Performance improvement

**Simulation Experimental Results.** The diverse task suites in the LIBERO benchmark reflect varying levels of instruction-following requirements from different perspectives. We conducted 500 trials for each task suite, and CogVLA achieved the highest success rate of 97.4%, as shown in

Table 4: **Ablation study on model components.** Pruning and TG-Pruning denote LFP-Routing w/o and w/ instruction guidance. All ablations maintain a fixed 8× overall sparsification ratio.

| Stage 1 | | Stage 2 | | Stage 3 | Spatial SR |
|---|---|---|---|---|---|
| Step 1 | Step 2 | Pruning | TG-Pruning | | |
| | | ✓ | ✓ | ✓ | 91.2 (-7.4) |
| | ✓ | ✓ | ✓ | ✓ | 96.0 (-2.6) |
| ✓ | | ✓ | ✓ | ✓ | 95.2 (-3.4) |
| ✓ | ✓ | | | ✓ | 92.0 (-6.6) |
| ✓ | ✓ | ✓ | | ✓ | 96.2 (-2.4) |
| ✓ | ✓ | ✓ | ✓ | | 92.0 (-6.6) |
| ✓ | ✓ | ✓ | ✓ | ✓ | **98.6** |

Table 5: **Ablation on sparsification ratio allocation.** Spf.Ratio denotes sparsification ratio, which can be tuned based on performance–efficiency trade-off.

| Stage 1 | Stage 2 | Spf.Ratio | Spatial SR |
|---|---|---|---|
| 1 × | 8 × | 8 × | 91.2 (-7.4) |
| 8 × | 1 × | 8 × | 92.0 (-6.6) |
| 2 × | 4 × | 8 × | 94.6 (-4.0) |
| 4 × | 2 × | 8 × | **98.6** |

Table 6: Comparison of Stage 1+2 with other visual compression methods.

| | FastV [12] | SliME [83] | Stage 1+2 |
|---|---|---|---|
| SR | 88.2 (-10.4) | 77.6 (-21.0) | **98.6** |

Tab. 1. This strong performance stems from CogVLA's 3-stage progressive design, which enhances instruction-driven perception throughout the manipulation process. Notably, CogVLA ranks second only in the LIBERO-Goal suite, primarily due to a deliberate trade-off between performance and efficiency—CogVLA reduces visual input by 8× compared to other VLA models in the table.

**Real-world Experimental Results.** We conducted real-world training and evaluation of the top four models from the LIBERO simulation benchmark on complex long-horizon tasks with rich instructions (Object Placement and Drawer Manipulation) and the representative dual-arm task T-shirt Folding. As shown in Tab. 2, CogVLA achieved the highest subtask and overall success rates. To better assess instruction-following ability in real-world settings, we collected ALOHA-based experimental data and applied data augmentation following LIBERO's protocol, including variations in spatial arrangements, manipulated objects, and their attributes. The results demonstrate that CogVLA's performance advantage generalizes effectively to real-world tasks.

### 3.3 Efficiency Optimization

As shown in Tab. 3, CogVLA achieves 2.79× faster inference time, 22.54× higher throughput, 3.12× lower FLOPs, and 2.49× reduction in training cost compared to OpenVLA. Moreover, CogVLA also outperforms state-of-the-art efficient VLA models such as OpenVLA-OFT and PD-VLA in both training and inference efficiency. These gains stem from: 1) instruction-driven vision sparsification achieved via EFA-Routing and LFP-Routing, reducing visual input by up to 8×; and 2) parallel action decoding enabled by bidirectional attention in the CAtten module.

### 3.4 Qualitative Analysis

As shown in Fig. 4, we compare the performance and efficiency of CogVLA and OpenVLA-OFT in both the LIBERO simulation and the ALOHA real-world setting. With strong instruction guidance and improved logical consistency, CogVLA executes manipulation tasks more accurately and avoids failures such as drawer collisions (e.g., row 3, column 5). CogVLA also demonstrates shorter action inference time, with its efficiency advantage becoming more pronounced as the task length increases.

### 3.5 Ablation Studies

Tab. 4 validates the effectiveness of each module within CogVLA's 3-stage progressive design, highlighting their synergistic contributions under a unified framework. Tab. 5 presents different sparsity ratio allocations across Stage 1 and Stage 2 under a fixed 8× vision sparsification. Results show that both stages contribute to performance gains, with larger Stage 1 ratios yielding greater improvements, as Stage 2 further filters instruction-relevant tokens based on Stage 1 outputs. Tab. 6 demonstrates that the combined Stage 1+2 sparsification outperforms existing methods, as it is more instruction-driven and deeply integrated into the overall CogVLA architecture. Additional ablation studies are provided in **Appendix C.3**.

## 4 Related Work

**Vision-Language Action (VLA) Models.** Vision-Language Models (VLMs) [35, 1, 42, 22, 57, 79, 81, 37, 68, 63, 86] have propelled robotic control by providing rich multimodal representations, fostering the development of VLA models [58, 43, 75, 67, 70, 56, 76, 6] that bridge perception and action generation. Early works like CLIPort [59] and PerAct [60] aligned visual features with language-conditioned action policies. The RT series [9, 8, 4] introduced action tokenization to enable scalable web-to-robot transfer. More recently, Octo [65] constructed a diverse multi-robot dataset to support multitask training, while OpenVLA [30] demonstrated superior generalization to household tasks compared to diffusion-based methods. The $\pi$ series [6, 27] proposed heterogeneous co-training across robots and semantic prediction tasks to enhance open-world generalization. However, directly fine-tuning billion-parameter VLMs for action prediction remains computationally intensive, limiting scalability.

**Efficient Design in VLA Models.** Improving VLA efficiency has largely followed two paths: LLM-centric and vision-centric. LLM-centric approaches include Mixture-of-Depth (MoD) pruning [51, 77, 46], dynamic reasoning depth adjustment [74, 82], sparse Mixture-of-Experts (MoE) architectures [16, 40, 11, 80], and lightweight models like DeeR-VLA [74], RoboMamb [44], and TinyVLA [69], all aiming to reduce decoding overhead. Vision-centric approaches focus on reducing the number of visual tokens passed to the LLM, employing techniques such as patch token selection based on similarity [55, 38], cropping-based techniques [45, 26], and an additional compression module [3, 39, 73]. However, naively adapting these methods often leads to semantic inconsistency across modalities due to a lack of unified sparsification. To address this, we propose a cognition-aligned, instruction-driven sparsification framework that jointly improves efficiency and cross-modal consistency.

## 5 Conclusion

We presented CogVLA, a cognition-aligned and instruction-driven Vision-Language-Action framework designed to address the computational inefficiencies and semantic fragmentation in existing VLA models. By integrating EFA-Routing, LFP-Routing, and CAtten into a unified 3-stage progressive design, CogVLA achieves effective vision sparsification and coherent cross-modal reasoning. Extensive evaluations on both the LIBERO benchmark and real-world robotic tasks demonstrate that CogVLA not only achieves state-of-the-art performance but also significantly reduces computational cost and inference latency. This work highlights the importance of instruction-driven multimodal sparsification in building scalable and efficient embodied AI systems.

## Acknowledgement

This study is supported by National Natural Science Foundation of China (Grant No. 62306090), Natural Science Foundation of Guangdong Province of China (Grant No. 2024A1515010147) and Shenzhen Science and Technology Program (KQTD20240729102207002).

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

# 🤖 CogVLA: Cognition-Aligned Vision-Language-Action Model via Instruction-Driven Routing & Sparsification

# Appendix

This appendix provides comprehensive supplementary material to support the methodology, analysis, and findings presented in the main paper.

- Section A describes implementation details, including model and training details.

- Section B outlines experimental details for both simulation and real-world settings.

- Section C presents extended quantitative analyses, including multi-seed evaluations, additional ablation studies, and expanded real-world results.

- Section D provides supplementary qualitative analyses, such as diverse task executions and instruction-to-observation attention visualizations.

- Section E discusses additional insights into the motivation behind CogVLA, highlights its current limitations, and reflects on the broader societal implications and potential risks.

- We provide third-person view videos at https://jiutian-vl.github.io/CogVLA-page , demonstrating CogVLA performing manipulation tasks in a fully autonomous mode, played at 1× speed. Due to the requirement of remote communication during each action chunk prediction, slight delays are introduced by network latency. For future deployments, we plan to run CogVLA locally on hardware with more than 20 GB of GPU memory (e.g., RTX 4090 with 24 GB) to eliminate such latency.

## A    Implementation Details

### A.1    Model Details

**EFA-Routing**. In Step 1, each of the two vision encoders uses 64 aggregation tokens, thereby reducing the number of visual tokens to 25% of the original. In addition, the scale and shift vectors for FiLM, $\gamma_i$ and $\beta_i$, are derived from a linear transformation of the text embedding. In Step 2, a two-layer MLP is applied to the text embedding to produce routing weights for the two vision encoders.

**LFP-Routing**. In this module, we employ a shifted cosine schedule [77] to control the proportion of visual tokens retained at each layer. The formulation is as follows:

$$\beta_l = \frac{1}{2} \cos \frac{\pi l}{L} + \eta, \quad l = 1, 2, \cdots, L \tag{21}$$

where $L$ denotes the total number of layers in the LLM, which is $L = 32$ for CogVLA. The constant $\eta$ is a shift factor that vertically adjusts the cosine decay curve, providing a flexible mechanism to control the overall computational cost of the model. In our implementation, $\eta$ is set to 0.5. Specifically, we apply a clamp operation to constrain $\beta_l$ within the range $[0.05, 0.85]$. As a result, LFP-Routing achieves approximately a 50% token pruning rate.

In addition, the instruction-conditioned scaling and shifting functions $\gamma_{\text{LLM}}(\cdot)$ and $\beta_{\text{LLM}}(\cdot)$ in LFP-Routing are both implemented using two-layer MLPs, with a hidden layer dimension of 2048, resulting in a parameter count almost identical to that of a direct linear layer.

### A.2    Training Details

**LIBERO Training Setup**. We adopt OpenVLA [31] as the backbone model and set the action chunk size to $K = 8$. Fine-tuning is performed using Low-Rank Adaptation (LoRA) with a rank of 32 and an $\alpha$ value of 64. The model is trained for 60K steps with a batch size of 64 and an initial learning rate of 5e-4. Checkpoints are evaluated every 10K steps, and the best-performing checkpoint is selected for reporting.

**Real-World Training Setup**. For the real-world experiments, we set the chunk size to $K = 25$ and fine-tune OpenVLA using LoRA with a rank of 32 and an alpha value of 64. The model is trained with a batch size of 32 for a total of 80K steps. The initial learning rate was set to 5e-4, which is reduced to 5e-5 after 50K steps. Starting from step 60K, we evaluate checkpoints every 10K steps and report the best-performing checkpoint.

# B  Experimental Details

## B.1  Simulation Benchmark

We evaluate CogVLA on the LIBERO simulation benchmark [41], a standardized suite of language-conditioned robotic manipulation tasks. Unlike earlier benchmarks such as RLBench [28], LIBERO features more complex and diverse instructions, averaging 10.48 words per command compared to only 3.34 in RLBench. This makes it a more suitable testbed for assessing the model's capacity in language grounding and multimodal reasoning. LIBERO comprises four task suites—**Spatial**, **Object**, **Goal**, and **Long**—each containing 10 tasks with 50 human-teleoperated demonstrations. These suites are designed to probe distinct reasoning capabilities:

- **LIBERO-Spatial** evaluates spatial reasoning capabilities by presenting identical objects arranged in different spatial configurations. The agent must interpret spatial relations (e.g., left/right, front/behind) described in the instruction to complete the task correctly.
- **LIBERO-Object** measures the model's ability to generalize across object categories. While spatial layouts remain fixed, the manipulated objects vary in type, shape, or color, requiring the agent to ground object-referential language and adapt its actions accordingly.
- **LIBERO-Goal** tests task-oriented comprehension by altering the goal specification while keeping object types and spatial layouts constant. The agent must disambiguate subtle differences in instruction semantics to execute distinct manipulation outcomes.
- **LIBERO-Long** challenges the agent with multi-step, long-horizon tasks involving diverse objects and environments. Success requires not only grounded perception and instruction following, but also sequential planning.

CogVLA is trained and evaluated under the same setting as OpenVLA [30] to ensure comparability. We report results on all four suites to validate the model's generalization, efficiency, and semantic grounding capabilities.

## B.2  Real-World Setup

We deploy CogVLA on Cobot Agilex ALOHA [20] manipulation platform, to validate its real-world applicability. The real-world evaluation consists of five diverse tasks involving both single-arm and coordinated dual-arm manipulation. To assess robustness and generalization, we introduce moderate data augmentation by varying object attributes (e.g., size, color) and rearranging spatial layouts.

We collect real-world training data for the Cobot Agilex ALOHA robot via human teleoperation. For Tasks 1–5, we gather 45, 45, 30, 30, and 45 expert demonstrations, respectively. We report the results of **Tasks 1–3** in the main paper, and provide additional results for **Tasks 4–5** in this appendix. The instructions and descriptions for **Tasks 1–5** are provided below:

- **Task 1**: *"Put the cube into the plate, and then put the toy into the bowl."*
  A two-step pick-and-place task involving object category understanding and temporal sequencing. This is a dual-arm task consisting of two sequential subtasks: *1) "Put the cube into the plate"* with the left arm, and *1) "Put the toy into the bowl"* with the right arm. Task success is achieved only when both subtasks are completed successfully. We report success rates for each subtask and the overall task.
- **Task 2**: *"Open the drawer, place the toy into the drawer, and then close it."*
  A composite task requiring interaction with articulated objects and multi-stage execution. This is a dual-arm task consisting of three sequential subtasks: *1) "Open the drawer"* with the left arm, *2) "Place the toy into the drawer"* with the right arm, and *3) "Close the drawer"* with the left arm. Task success requires all three subtasks to be completed. We report success rates for each subtask and the overall task.

Table 7: **Multi-seed evaluation results in simulation.** Task success rates (SR) are compared across four task categories on the LIBERO benchmark. "†" denotes our reproduced results. CogVLA demonstrates strong and consistent performance.

| Method | Spatial SR ↑ | Object SR ↑ | Goal SR ↑ | Long SR ↑ | Average SR ↑ | RK ↓ |
|---|---|---|---|---|---|---|
| OpenVLA *[CoRL'24]* [30] | $84.7 \pm 0.9$ | $88.4 \pm 0.8$ | $79.2 \pm 1.0$ | $53.7 \pm 1.3$ | $76.5 \pm 0.6$ | 5 |
| SpatialVLA *[RSS'25]* [50] | $88.2 \pm 0.5$ | $89.9 \pm 0.7$ | $78.6 \pm 0.6$ | $55.5 \pm 1.0$ | $78.1 \pm 0.7$ | 4 |
| STAR *[ICML'25]* [23] | $95.5 \pm 0.6$ | $98.3 \pm 0.2$ | $95.0 \pm 0.7$ | $88.5 \pm 0.3$ | $94.3 \pm 0.1$ | 2 |
| CoT-VLA *[CVPR'25]* [84] | $87.5 \pm 1.4$ | $91.6 \pm 0.5$ | $87.6 \pm 0.6$ | $69.0 \pm 0.8$ | $83.9 \pm 0.6$ | 3 |
| CogVLA | $\mathbf{98.5 \pm 0.5}$ | $\mathbf{98.8 \pm 0.4}$ | $\mathbf{96.5 \pm 0.6}$ | $\mathbf{95.2 \pm 1.1}$ | $\mathbf{97.4 \pm 0.4}$ | **1** |

- **Task 3**: *"Fold the T-shirt."*
  A soft-body manipulation task that evaluates the system's ability to handle deformable objects. This is a dual-arm task consisting of three sequential folding steps. Task success is determined by the successful execution of all three steps. We report intermediate success rates for each step and the overall task performance.

- **Task 4**: *"Pick the red cube into the plate, and then pick the big cube into the bowl."*
  A multi-attribute grounding task requiring comprehension of both color and size references. This is a dual-arm task consisting of two sequential subtasks: *1) "Pick the red cube into the plate"* with the left arm, and *2) "Pick the big cube into the bowl"* with the right arm. Task success is achieved only when both subtasks are completed. We report success rates for each subtask and the overall task.

- **Task 5**: *"Pick the left cube into the plate."*
  A spatial reasoning task focusing on relative positioning and egocentric understanding. This is a single-arm task consisting of one pick-and-place action. We report the final task success rate.

## C  Supplementary Quantitative Analysis

### C.1  Multi-Seed Evaluation

To evaluate the statistical robustness and consistency of CogVLA's performance, we conduct multi-seed evaluations on the LIBERO benchmark. For each of the four task suites (Spatial, Object, Goal, and Long), we run experiments using three independent random seeds and report the mean success rate along with the standard deviation.

As shown in **Tab. 7**, CogVLA exhibits consistently high performance across different seeds, with standard deviations ranging from 0.2% to 0.6%. This indicates stable learning behavior and further validates the strong generalization capability of CogVLA's three-stage instruction-driven architecture across diverse task types.

### C.2  Extended Real-World Task Results

In addition to the results reported in the main paper, we present the performance of CogVLA on Tasks 4 and 5, as shown in **Tab. 8**.

- **Task 4** (*"Pick the red cube into the plate, and then pick the big cube into the bowl"*) evaluates the model's ability to ground multi-attribute language and execute sequential actions. CogVLA achieves the highest success rates across both subtasks and the overall task, demonstrating strong compositional understanding of attribute references such as color and size.

- **Task 5** (*"Pick the left cube into the plate"*) focuses on egocentric spatial reasoning, requiring precise interpretation of relative spatial references from the agent's visual perspective. CogVLA maintains a high success rate in this setting, indicating robust grounding of spatial concepts.

Table 8: **Extended real-world results on Tasks 4–5.** Performance comparison on the Cobot Agilex ALOHA tasks. "†" indicates our reproduced results.

| Method | Task 4 | | Task 5 | Average |
|--------|--------|--|--------|---------|
| | Red Cube→Plate | +Big Cube→Bowl | Left Cube→Plate | SR |
| PD-VLA† [62] | 7/10 | 5/10 | 6/10 | 60.0% |
| OpenVLA-OFT† [29] | 7/10 | 6/10 | 6/10 | 63.3% |
| CogVLA | **8/10** | **7/10** | **8/10** | **76.7%** |

These results further validate CogVLA's ability to generalize to real-world tasks that demand fine-grained language grounding and spatial understanding.

## C.3 Extended Ablation Studies

We extend the sparsification analysis by evaluating additional Stage 1/Stage 2 configurations: **2×2** and **4×4**, while keeping the total sparsification ratio fixed at 4× and 16×, respectively. These configurations are compared alongside the baseline 8× setting with different asymmetric allocations (e.g., 2×–4× and 4×–2×), allowing us to systematically assess how the distribution of sparsity across stages impacts downstream performance.

As shown in **Tab.3**, the **2×2** setting provides a favorable trade-off between performance and computational efficiency. In contrast, the **4×4** setting leads to a slight degradation in performance, suggesting that excessive sparsification across both stages may hinder the preservation of task-relevant information.

Interestingly, the asymmetric configurations, particularly the **4×–2×** setup, outperform their symmetric counterparts, achieving the highest spatial success rate of 98.6. This highlights the advantage of applying a more aggressive token reduction in Stage 1 (EFA-Routing), where redundant visual tokens can be effectively compressed via instruction-guided aggregation. Subsequently, Stage 2 (LFP-Routing) performs finer-grained token pruning in a context-aware manner within the language model, allowing for better preservation of task-relevant information.

Table 9: **Supplementary ablation on sparsification ratio allocation.** Spf.Ratio denotes the sparsification ratio, which can be adjusted based on the performance–efficiency trade-off. CogVLA achieves better performance when a relatively higher sparsification ratio is allocated to Stage 1 compared to Stage 2.

| Stage 1 | Stage 2 | Spf.Ratio | Spatial SR | FLOPs |
|---------|---------|-----------|------------|-------|
| 2 × | 2 × | 4 × | 96.4 (-2.2) | 3.87 T |
| 4 × | 4 × | 16 × | 93.2 (-5.4) | 2.30 T |
| 2 × | 4 × | 8 × | 94.6 (-4.0) | 2.72 T |
| 4 × | 2 × | 8 × | **98.6** | 2.72 T |

These findings support the core design principle of CogVLA: progressive sparsification with an asymmetric allocation tailored to the representational characteristics of each stage. By balancing early-stage compression and late-stage selectivity, the model achieves both computational efficiency and high task accuracy, reinforcing the importance of stage-aware sparsity scheduling in multimodal architectures.

## D Supplementary Qualitative Analysis

### D.1 Additional Visualizations of Simulation and Real-World Results

We present additional qualitative results from both simulation and real-world experiments to illustrate CogVLA's generalization and execution capabilities. As shown in **Fig. 8**, the model consistently completes multi-step tasks across diverse environments, object configurations, and instruction variants.

In real-world tasks with varying instructions, CogVLA accurately interprets long-horizon commands and produces coherent action sequences. These examples further highlight the model's ability to maintain cross-modal consistency and temporal reasoning, as well as its robustness in simulation-to-reality transfer. **Fig. 5** illustrates the real-world manipulation workflows for Tasks 1-5. For Task 1, we provide multi-view observations from the *Front Camera*, *Left Wrist Camera*, and *Right Wrist*

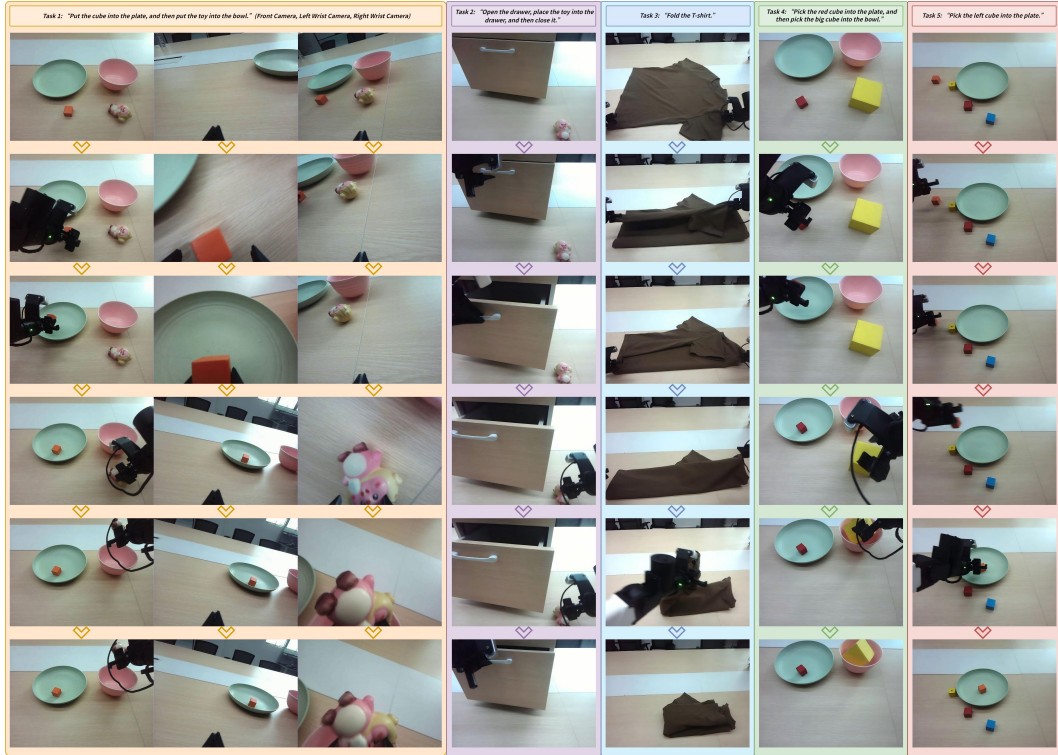

Figure 5: **Real-world Manipulation Workflows and Visualizations for Tasks 1–5.** Each task panel illustrates the initial setup and CogVLA's execution process based on the natural language instruction. For Task 1, multi-view observations from the *Front Camera*, *Left Wrist Camera*, and *Right Wrist Camera* are provided to capture dual-arm coordination. For Tasks 2–5, representative frames from the *Front Camera* highlight key manipulation stages. These visualizations support interpretation of task complexity and grounding behavior.

*Camera.* For Tasks 2-5, only *Front Camera* observations are shown for clarity. In **Fig. 6**, we present a third-person view demonstration of CogVLA performing a manipulation task in the lab. The corresponding MP4 video file is provided in the supplementary materials.

### D.2 Instruction-to-Observation Attention Maps

To gain deeper insights into how CogVLA aligns language instructions with visual observations, we visualize the attention maps generated by the cross-modal attention modules. As shown in **Fig. 7**, the attention weights highlight task-relevant regions in the input image.

These visualizations demonstrate that CogVLA's instruction-aware routing mechanisms effectively guide the perception module to attend to semantically meaningful areas, enabling robust visual grounding even in cluttered or ambiguous scenes.

## E Discussion

### E.1 Supplementary Details on the Motivation

CogVLA is motivated by the need to improve both computational efficiency and cross-modal semantic alignment in instruction-conditioned robotic systems. Its architectural design is informed by cognitive science research on how humans process language, perceive their environment, and execute actions in a coherent and goal-directed manner.

Cognitive studies suggest that humans rely on structured inductive biases—often termed "intuitive theories"—to interpret the world, including intuitive physics, causality, and theory of mind [33, 66].

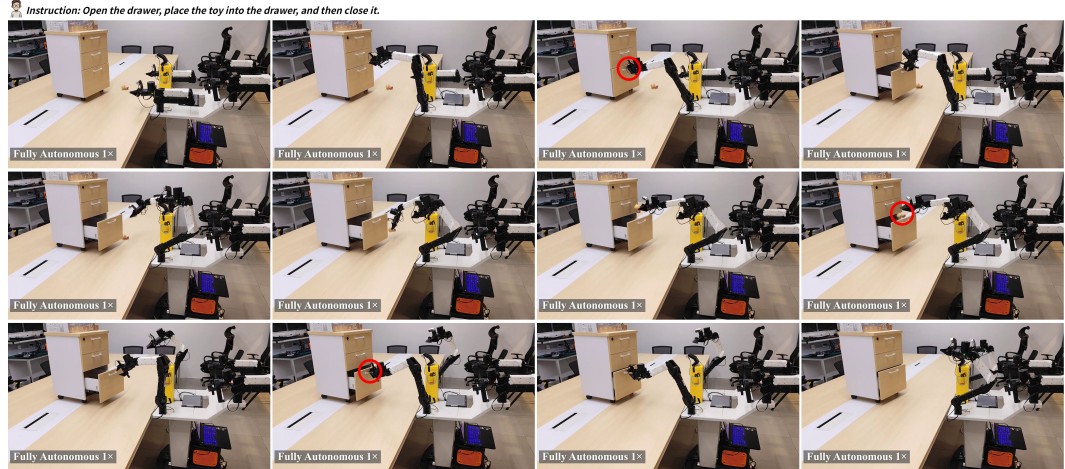

Figure 6: **Third-person visualization of CogVLA performing a manipulation task.** The corresponding video is provided in the supplementary materials. Gripper details are highlighted with red circles.

While recent multimodal large language models exhibit partial competence in these areas, they often lack robustness in compositional reasoning and causally grounded behavior [53].

To address these limitations, CogVLA adopts a biologically inspired architecture that reflects the division of functional roles observed in the human brain. Specifically, we draw connections between the model's three routing modules and key components in human multimodal cognition: the **Visual Attention System (VAS)**, the **Supplementary Motor Area (SMA)**, and the **Premotor Cortex (PMC)**.

**Visual Attention System (VAS) → Encoder-FiLM.** The human visual attention system selectively enhances perception of task-relevant features while suppressing distractors [15]. Top-down signals from frontal and parietal cortices bias visual processing toward objects or regions mentioned in language or necessary for action. This selective modulation improves efficiency and semantic grounding in complex scenes. In CogVLA, the **Encoder-FiLM** module mimics VAS by dynamically modulating visual encoder features conditioned on instructions, focusing perception on semantically relevant regions and reducing redundancy [48]. This allows the model's perception to be grounded in context, much as the brain's attention system tunes visual processing to relevant aspects of a scene during coordinated vision-language tasks.

**Supplementary Motor Area (SMA) → LLM-FiLM.** The SMA plays a key role in planning and sequencing actions, even in the absence of physical movement [54, 64]. It integrates multimodal information and high-level goals to shape future motor behavior, before engaging primary motor circuits. In CogVLA, the **LLM-FiLM** module can be seen as the "intention planner" of the model and serves a similar function: it injects task-specific intent into the language model, pruning irrelevant visual-linguistic tokens and steering the model toward generating appropriate action plans. This enables more efficient and intention-aligned reasoning, analogous to how the SMA organizes abstract motor programs before execution.

**Premotor Cortex (PMC) → V-L-A Coupled Attention.** The premotor cortex is involved in translating perceptual cues into executable motor plans [21, 19]. It contains visuomotor neurons that represent both the perception of object affordances and the intended grasping actions, enabling visuomotor grounding. CogVLA's **V-L-A Coupled Attention** module reflects this mechanism by integrating visual, linguistic, and action representations through a unified attention mechanism. This ensures that generated actions are causally and temporally coherent with respect to both the observed scene and the given instruction.

By aligning its modular design with biologically plausible cognitive functions, CogVLA offers not only performance and efficiency gains, but also a cognitively grounded pathway for improving generalization and interpretability in embodied multimodal agents.

👤 *Instruction: Pick up the black bowl between the plate and the ramekin and place it on the plate.*

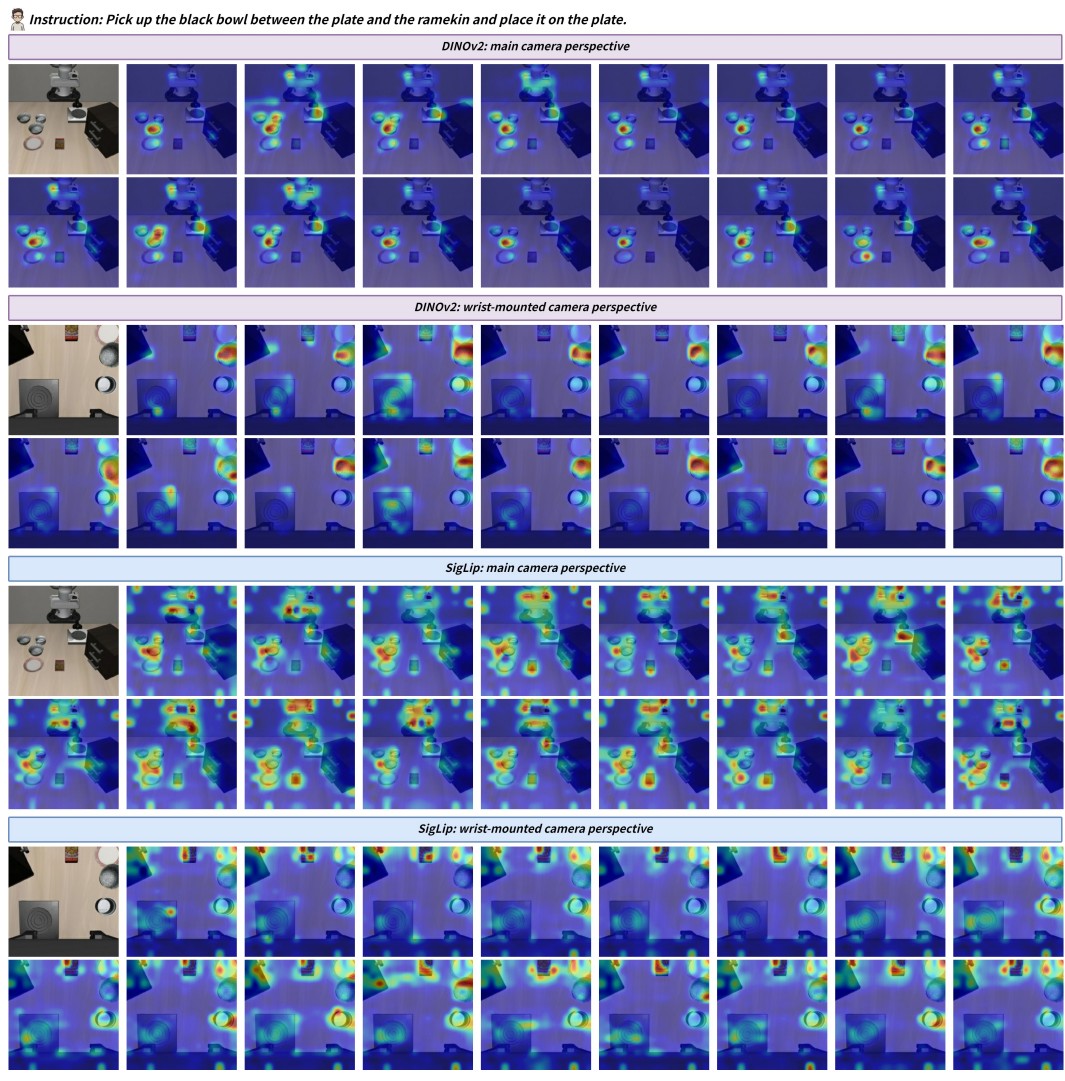

Figure 7: **Attention maps between aggregation tokens and patch tokens in DINOv2 and SigLIP.** We visualize the attention maps from 17 out of 64 aggregation tokens to the patch tokens of the observation, covering four sets of visualizations across two visual encoders and two camera views. The input language instruction is: "Pick up the black bowl between the plate and the ramekin and place it on the plate." Both DINOv2 and SigLIP exhibit varying degrees of focused attention on patch tokens relevant to the instruction.

## E.2 Limitation and Future Work

While CogVLA demonstrates strong performance across simulation and real-world tasks, several limitations remain. First, the current instruction-to-vision routing relies on predefined sparsity ratios and fixed token pruning schedules, which may not adapt optimally to varying instruction complexity or scene difficulty. Second, although the model generalizes well within the LIBERO and ALOHA settings, its performance under out-of-distribution instructions or unseen manipulation categories is yet to be thoroughly evaluated.

In future work, we aim to explore adaptive sparsification mechanisms conditioned on task semantics and environmental uncertainty. Moreover, integrating lifelong learning or online adaptation strategies may further enhance CogVLA's robustness in open-world deployment scenarios. Lastly, extending the framework to support multimodal feedback (e.g., haptic or force sensing) could improve its applicability to fine-grained manipulation tasks.

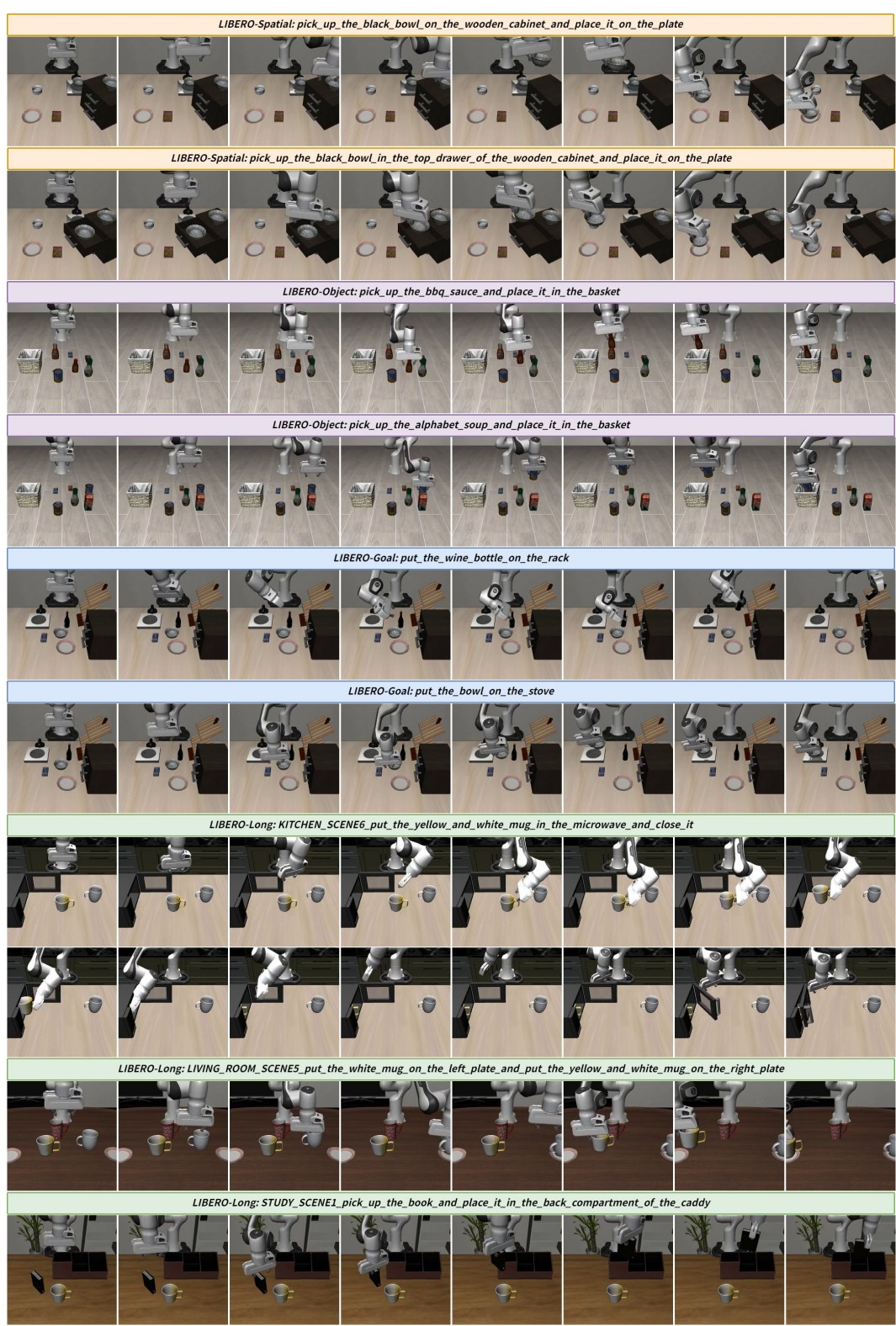

Figure 8: **Manipulation Workflows and Visualizations in the LIBERO Simulation Environment.** We present the execution processes of CogVLA across LIBERO-Spatial, LIBERO-Object, LIBERO-Goal, and LIBERO-Long, demonstrating its strong performance under diverse instructions and a wide range of tasks.

### E.3 Broader Impact and Potential Risk

CogVLA advances the efficiency and interpretability of instruction-driven robotic manipulation, offering potential benefits in applications such as assistive robotics, household automation, and industrial assembly. Its biologically inspired sparsification and routing mechanisms reduce computation cost, making it more accessible for resource-constrained platforms. However, as with any vision-language-action system, risks include misinterpretation of ambiguous instructions, failure in unpredictable environments, and bias amplification from training data. If deployed in safety-critical settings without appropriate safeguards, such failures could lead to unintended behaviors or physical harm. We encourage the community to adopt robust evaluation protocols, prioritize transparency in model behavior, and consider human-in-the-loop designs to mitigate such risks. Broader societal considerations—including data diversity, accessibility, and responsible deployment—should guide future development of systems built upon CogVLA.

