# OpenReview forum: "CogVLA: Cognition-Aligned Vision-Language-Action Models via Instruction-Driven Routing & Sparsification"
_NeurIPS.cc/2025/Conference — NeurIPS 2025 poster_

### Official Review · Reviewer_EsjF · 2025-06-22

**Clarity:** 3
**Significance:** 2
**Originality:** 2
**Rating:** 4
**Confidence:** 4

**Summary:**

The paper introduce CogVLA, a Cognition-Aligned Vision-Language-Action framework that leverages instruction-driven routing and sparsification to improve both efficiency and performance. Specifically, it contains of three modules and innovations: 1) Encoder-FiLM based Aggregation Routing, 2)LLM-FiLM based Pruning Routing and V‑L‑A Coupled Attention.

**Questions:**

See Above.

**Ethical Concerns:**

["NO or VERY MINOR ethics concerns only"]

**Final Justification:**

After reviewing the author's rebuttal, I decide to maintain my score which still holds an up-word rating about this paper with borderline accept.

**Limitations:**

Yes

**Quality:**

3

**Strengths And Weaknesses:**

Strengths:
1. This paper is clearly written.
2. It proposes a combination approach by incorporation of Encoder-FiLM based Aggregation Routing, LLM-FiLM based Pruning Routing and a novel V‑L‑A Coupled Attention for VLA framework.
3. Experimental results on LIBERO and real-world tasks demonstrate its effectiveness, while also being efficient.

Weakness:
1. Despite noted as Cognition-Aligned VLA model, I find it hard to synergy author's three approachs into a coherent CogVLA framework.
2. The method is more combinative and separated contributions instead of totally novel. For example, the use aggregating different visual features is an intuitive method by various approach needs multiple level feature map such as Depth Transformer. The attention mask approach is also very straightforward.
3. How do this model compare with large scale trained models such as Pi series and Gemini Robotics? The author may note the scale of training compute differences and capability differences.

---

> ### Author Rebuttal · Authors · 2025-07-30
>
> We thank you for your positive recognition of the paper's structure, clarity, and methodological effectiveness, as well as the thoughtful and critical insights on CogVLA. Below is our detailed response to the identified concern:
>
> ### W1: Despite noted as ... framework.
>
> ### R1: Clarifying the cognition-aligned architecture through perception–intention–reasoning mapping.
>
> > The architecture of CogVLA is not a simple combination of three techniques. Rather, it is systematically constructed based on the division of labor in human multimodal cognition—specifically the perception–intention–reasoning pathway. **The three modules in CogVLA have clearly defined functional correspondences and semantic dependencies**, as shown below:
> >
> > | Human Cognitive Pathway| CogVLA Module   | Functional Stage       |
> > | ------------------------------ | --------------- | ---------------------- |
> > | VAS (Visual Attention System)  | **EFA-Routing** | **Perceptual Aggregation** |
> > | SMA (Supplementary Motor Area) | **LFP-Routing** | **Intention Construction** |
> > | PMC (Premotor Cortex)          | **CAtten**      | **Reasoning & Execution**  |
> >
> > We will include this mapping diagram in the revised version to clarify the cooperative structure among modules. Detailed explanations are as follows:
> >
> > - **Module Synergy**:
> >      **EFA-Routing emulates the VAS** by aggregating task-relevant visual features through instruction-guided modulation; **LFP-Routing corresponds to the SMA**, injecting linguistic intent into perceptual representations and pruning redundant tokens to form executable motor plans; **CAtten mirrors the PMC**, integrating linguistic reasoning and action decoding within a shared context to ensure coherent and semantically aligned action generation. **These three components collaboratively form a brain-inspired perception–intention–execution pipeline.**
> > - **Information Flow**:
> >      **CogVLA adopts a strictly cascaded structure**: tokens selected by EFA-Routing serve as the sole input to LFP-Routing, and the pruned outputs from LFP-Routing are the exclusive decoding source for CAtten. This ensures a unified and consistent architecture, rather than a plug-and-play combination of independent methods.
> >
> > In essence, the **EFA-Routing → LFP-Routing → CAtten** flow in CogVLA corresponds directly to the human **VAS → SMA → PMC** pathway. This end-to-end cognitive alignment is the defining characteristic of our Cognition-Aligned VLA framework.
> >
>
> ------
>
> ------
>
> ### W2: The method is more ... straightforward.
>
> ### R2: Explaining the integrative innovations behind aggregation, pruning, and attention mechanisms.
>
> > As discussed in R1, CogVLA is not a naïve combination of existing techniques. Its advantage arises from **a cognitively aligned, three-stage collaborative design**. While we acknowledge that some sub-functionalities build upon established prior work, our specific module configurations, architectural composition, and control logic are novel. We elaborate on the following key innovations:
> >
> > **(1) Aggregation of Visual Features**
> >
> > - **Motivational Innovation**: Instruction-aligned token aggregation.
> >      Our goal in visual feature aggregation is not merely to sparsify the token set, but to retain tokens that are semantically aligned with task instructions. **As visualized in Appendix D.2, the retained 1/8 of visual tokens preserve both local object anchors and global action-relevant cues, demonstrating semantically meaningful sparsification.**
> > - **Design Innovation**: Progressive, intra- and inter-encoder aggregation.
> >      EFA-Routing conducts visual token aggregation **in two stages**—first within a single visual encoder, then across two different encoders. **The intra-encoder aggregation follows an autoregressive structure, while inter-encoder fusion is guided by adaptive weighting, enabling complementary feature integration across distinct encoding branches.**
> >
> > **(2) Attention Mask Strategy**
> >
> > - **Motivational and Design Innovation**: Joint improvement of reasoning ability and inference efficiency.
> >      In VLA tasks, attention windows span `[observation-instruction-action]`, where action tokens possess unique temporal and semantic roles. Our CAtten module adopts a tailored attention design:
> >     - Causal autoregression within `[observation]` and `[instruction]` preserves the reasoning capacity inherited from pretrained LLMs.
> >     - Bidirectional attention within `[action]` allows parallel decoding, enhancing sequence continuity and inference efficiency.
> >     - Task intent is injected from `[instruction]` to `[observation]` via LLM-FiLM, avoiding the performance degradation seen in $\pi_0$ and OpenVLA-OFT caused by full bidirectional attention across `[observation, instruction]`.
> >
> > **(3) Unified Cognition-Aligned Framework**
> >
> > - We believe CogVLA’s core contribution lies in realizing a cognition-aligned architecture that is not only theoretically grounded through explicit cognitive mappings, but also empirically validated. It achieves **a triple advantage in performance, efficiency, and generalization**—which would be difficult to attain through a simple additive combination of components.
> >
> > The idea of modeling robotic manipulation based on human cognitive and motor processes is both promising and impactful, opening avenues for deeper understanding of embodied intelligence.
>
> ------
>
> ------
>
> ### W3: How do this model ... differences.
>
> ### R3: Comparing training efficiency and long-horizon performance with $\pi$ series and Gemini Robotics.
>
> > We thank the reviewer for raising this important yet previously under-discussed dimension. Below, we compare CogVLA with the $\pi$ series and Gemini Robotics in terms of both training compute and model capability.
> >
> > **(1) $\pi$ Series**
> >
> > - **Training Compute**:
> >
> >     Compared to the $\pi$ series, CogVLA employs a larger LLM backbone but still achieves superior training efficiency, significantly reducing GPU time and computational scale, as shown below:
> >
> >     | Method       | LLM Backbone | Training Steps | Training Time (per 10k) | Relative Training Cost |
> >    | ------------ | ------------ | -------------- | ----------------------- | ---------------------- |
> >     | $\pi_0$      | 2.6B         | 800–1200k      | 4.3 h                   | **~14.1×**             |
> >     | $\pi_0$+FAST | 2.6B         | 80–200k        | 6.3 h                   | **~2.9×**              |
> >     | $\pi_{0.5}$  | 2.6B         | -              | -                       | -                      |
> >     | OpenVLA-OFT  | 7B           | 50–150k        | 12.5 h                  | **~4.0×**              |
> >     | **CogVLA**   | **7B**       | **50–80k**     | **4.7 h**               | **1 (baseline)**       |
> >
> >     Notes:
> >     (i) We reproduced the $\pi$ series based on the open-source `openpi` codebase.
> >      (ii) All models were trained on 4× A800 (80GB) GPUs.
> >      (iii) Due to the unavailability of $\pi_{0.5}$ code, we could not include its training evaluation.
> >
> > - **Model Capability**:
> >
> >     Beyond training efficiency, CogVLA also demonstrates strong capability—especially on long-horizon tasks such as LIBERO-Long—where its performance is substantially better than the $\pi$ series:
> >
> >     | Method                               | LIBERO-Spatial | LIBERO-Object | LIBERO-Goal | LIBERO-Long      | LIBERO-Overall |
> >    | ------------------------------------ | -------------- | ------------- | ----------- | ---------------- | -------------- |
> >     | $\pi_0$                              | 96.8           | **98.8**      | 95.8        | 85.2 **(−10.2)** | 94.2           |
> >     | $\pi_0$+FAST                         | 96.4           | 96.8          | 88.6        | 60.2 **(−35.4)** | 85.5           |
> >     | $\pi_{0.5}$  (from scratch)          | 96.6           | 97.2          | 94.6        | 84.8 **(−10.4)** | 92.7           |
> >     | $\pi_{0.5}$  (from generalist model) | 98.0           | 97.8          | 95.6        | 85.8 **(−9.4)**  | 96.0           |
> >     | **CogVLA**                           | **98.6**       | **98.8**      | **96.6**    | **95.4**         | **97.4**       |
> >
> > - We will include this comparative analysis in the main table of our revised submission to more comprehensively showcase CogVLA’s advantages over state-of-the-art baselines.
> >
> > **(2) Gemini Robotics**
> >
> > - Gemini Robotics is not open-sourced and primarily evaluated on the internal Gemini Robotics Benchmark defined by Google DeepMind, which we currently cannot access or reproduce. Therefore, to provide a meaningful comparison, we followed the task definitions from the Gemini paper and implemented them on the ALOHA platform. CogVLA achieves higher accuracy, particularly on tasks where Gemini Robotics shows suboptimal performance:
> >
> >     |                 | Task 1   | Task 2   | Task 3    |
> >    | --------------- | -------- | -------- | --------- |
> >     | Gemini Robotics | 2/10     | 5/10     | 10/10     |
> >     | $\pi_0$         | 0/10     | 2/10     | 9/10      |
> >     | **CogVLA**      | **4/10** | **6/10** | **10/10** |
> >
> >     Task definitions: Task 1 – Unfold the mat; Task 2 – Fold the blue T-shirt from bottom to top; Task 3 – Close the laptop.
> >
> > - Additional experiments will be conducted upon receiving access via Google Trusted Tester evaluation, and the results will be incorporated in future updates.
>
> We sincerely thank you for the insightful suggestions, which have been invaluable in refining the paper’s structure and empirical support. We hope our responses sufficiently address your concerns.

---

> > ### Author Response · Authors · 2025-08-04
> > **Invitation for Further Discussion**
> >
> > Dear Reviewer EsjF,
> >
> > Thank you again for your valuable comments. We hope that our clarifications and additional experiments in the rebuttal have addressed each of your concerns. Should any questions remain unclear, we would appreciate the opportunity for further discussion.
> >
> > Best regards, Authors

---

> > > ### Comment · Reviewer_EsjF · 2025-08-05
> > >
> > > Thank the authors for the detailed rebuttal, I maintain my positive score about this paper.

---

> > > > ### Author Response · Authors · 2025-08-05
> > > >
> > > > Dear Reviewer EsjF,
> > > >
> > > > Thank you very much for the time and effort you dedicated to reviewing our rebuttal. We would sincerely welcome the opportunity to engage in further discussion.
> > > >
> > > > Best regards, Authors

---

### Official Review · Reviewer_Bu5w · 2025-06-23

**Clarity:** 2
**Significance:** 3
**Originality:** 2
**Rating:** 4
**Confidence:** 4

**Summary:**

This paper proposes a novel instruction-driven VLA framework, CogVLA, which achieves efficient cross-modal sparsification and semantic alignment through a three-stage progressive design (EFA-Routing, LFP-Routing, CAtten). The framework adopts a biologically inspired architecture that simulates human multimodal coordination mechanisms to enhance the coherence of perception-reasoning-control. The method achieves SOTA performance on the LIBERO benchmark and real-robot tasks, while significantly reducing training costs and inference latency.

**Questions:**

1. I hope the authors will carefully consider the first **Major Weakness**. While the paper itself cannot be changed during review, I would be willing to increase my score if the rebuttal addresses this concern thoughtfully.

2. I would like the authors to supplement their results with evaluations on additional VLA datasets and include GPU-wise comparisons for the reported efficiency gains.

3. I would appreciate it if the authors could specify how they plan to improve the clarity of figures and tables in future versions of the paper.

**Ethical Concerns:**

["NO or VERY MINOR ethics concerns only"]

**Final Justification:**

I raise my score to 4： Borderline accept. Some limites and errors should be refined in future  versions.

**Limitations:**

yes

**Quality:**

3

**Strengths And Weaknesses:**

## Strengths

- **Significant gains in both efficiency and performance**:
  The method achieves a success rate of 97.4% on LIBERO and 70.0% on real-robot tasks. Compared to OpenVLA, it reduces training cost by 2.49× and inference latency by 2.79×, making it a rare case of simultaneous improvement in both speed and effectiveness.

- **Original method design**:
  The three-stage architecture and cognitive alignment modules are innovative and well-supported by extensive ablation studies.

- **Clear writing and detailed supplementary materials**:
  The paper is well-structured and accessible, which facilitates reader understanding.

---

## Weaknesses

### Major Weaknesses

1. **Insufficient integration of biological motivation and experimental validation**:
   The three-stage cognitive alignment design of CogVLA is framed using biological terminology (VAS, SMA, PMC, etc.) inspired by human cognitive processes. However, the paper lacks either of the following:

   - **Solid support from cognitive science theory**:
     While the paper cites relevant literature (e.g., references [24], [43]), it does not clearly explain how these biological concepts are incorporated into the VLA design. As a result, the biological analogy feels more like a conceptual metaphor than a theoretically grounded design principle.

   - **Experimental validation of the biological motivation**:
     A compelling way to justify such biologically inspired design would be through empirical results—e.g., showing that CogVLA achieves performance that mirrors certain aspects of human behavior, or that its design leads to improvements specifically in tasks known to benefit from human-like reasoning. Unfortunately, the current version of the paper does not include such comparative analyses, either quantitatively or qualitatively (e.g., visualizations).

2. **Limited comparative experiments**:
   The evaluation in simulation is limited to the LIBERO dataset. Although LIBERO is widely used, the performance gap between CogVLA and OpenVLA-OFT is relatively small. Evaluating on additional VLA benchmarks (e.g., BridgeData V2, CALVIN) would better demonstrate the generality and robustness of CogVLA.
In Table 4, the results seem reveal that the stage 3 is critical with improvement of 6.6%. It is hard to understand why the CAtten can bring such considerable improvements. Also the other results indicate that each component is fluctuant.


3. **Lack of hardware variation in efficiency results**:
   The **Efficiency Optimization Results** section lacks comparative analysis across different GPUs. Since reduced training cost and inference latency are claimed as major contributions, hardware configuration becomes a critical factor and should be considered explicitly. This paper also mentions the other speed up methods, such as MoD. However, it does not compare with these methods.


---

### Minor Weaknesses

- **Generalization remains underexplored**:
  As acknowledged by the authors, “its performance under out-of-distribution instructions or unseen manipulation categories is yet to be thoroughly evaluated.” Including such results would further support the biologically inspired design's validity, especially for tasks humans perform naturally but current SOTA methods often fail.

- **Missing explanations in figures**:
  Some key diagrams (e.g., Figures 2 and 3) lack detailed annotations for arrows and symbols, which hinders reader comprehension. Figure 1 is too compact to see the details clearly. In section 2, the “I”and “t”in eqation 1 are not be defined.

- **Typographical error**:
  In Table 3, “Taining Cost” should be corrected to “Training Cost”.

---

> ### Author Rebuttal · Authors · 2025-07-30
>
> Thank you for the detailed and constructive feedback. We appreciate your recognition of our contributions and welcome the valuable suggestions for improvement.
>
> ### W1: Insufficient integration... validation.
>
> ### R1: We address this concern through both theoretical grounding and experimental validation.
>
> **(1) Theoretical Foundation: Cognitive Science-Inspired Architecture**
>
> |Index|Human→Robot|Stage|
> |-|-|-|
> |(1.1)|VAS→EFA-Routing|Perception|
> |(1.2)|SMA→LFP-Routing|Intention|
> |(1.3)|PMC→CAtten|Reasoning|
>
> We will add a figure illustrating the mapping from human cognition **(VAS → SMA → PMC)** to CogVLA modules **(EFA-Routing → LFP-Routing → CAtten)** in the main paper.
>
> > **(1.1) VAS → EFA-Routing**
> >
> >
> > In the human brain, VAS selectively enhances perception of task-relevant features through top-down signals from the prefrontal and parietal cortex, while suppressing distractors. This mechanism improves semantic alignment and efficiency in visuomotor execution, especially under complex scenes. **To emulate this, EFA-Routing:**
> >
> > - Uses Encoder-FiLM to modulate visual encoding based on **instruction semantics**.
> > - Aggregates 25% of patch tokens via autoregressive attention, **reducing redundancy**.
> > - Combines local (DINOv2) and global (SigLIP) features via an Aggregation Router, enabling **spatial precision and semantic grounding**.
> >
> > **We visualize its token selection via heatmaps in Appendix D.2, demonstrating its effectiveness in implementing VAS-like behavior.**
> >
> > ------
> >
> > **(1.2) SMA→LFP-Routing**
> >
> > SMA is critical for multi-step planning and intention modeling. It integrates multimodal context and high-level goals, often activating even before actual action, leveraging prior knowledge to form action intentions before execution. **To emulate such prior intent construction, LFP-Routing:**
> >
> > - Uses LLM-FiLM to **inject instruction-driven intent** into observation tokens within the shared LLM context without disrupting LLM reasoning (avoiding bidirectional attention across modalities).
> > - Prunes 50% of observation tokens via action supervision, **improving alignment and efficiency** while preserving reasoning fidelity.
> >
> > ------
> >
> > **(1.3) PMC→CAtten**
> >
> > PMC generates executable, fluent motor plans based on current observations. It supports high-level reasoning during transitions (e.g., from “pick” to “place”), while allowing low-level motion execution to proceed smoothly. **CAtten mimics PMC by combining two complementary attention mechanisms:**
> >
> > - Causal Vision-Language Attention for reasoning and intent alignment;
> > - Bidirectional Action Chunk Attention for fluent and efficient decoding.
> >
> > These are integrated in a Unified Hybrid Attention within a shared context window, achieving a balance between reasoning and real-time execution.
> >
>
> CogVLA’s **EFA-Routing**, **LFP-Routing**, and **V-L-A Coupled Attention** reflect the roles of VAS, SMA, and PMC in human cognition. We believe modeling human cognitive pathways offers a promising direction for embodied intelligence.
>
> **(2) Empirical Validation of Cognitive Design**
>
> > **(2.1) Quantitative Evaluation on Cognition-Level Tasks**
> >
> > VLABench is designed to evaluate **world knowledge and commonsense reasoning** through implicit, non-templated instructions, making it ideal for testing cognition-level capabilities. Compared with official baselines, **CogVLA achieves significant absolute gains across all tasks**:
> >
> > |Method|Select Book|Select Drink|Select Fruit|Select Tube|Add Condiment|Insert Flower|Overall|
> > |-|-|-|-|-|-|-|-|
> > |Octo|0.0000|0.0000|0.0000|0.0154|0.0154|0.0000|0.0051|
> > |OpenVLA|0.0769|0.0846|0.0462|0.0769|0.1238|0.1385|0.0911|
> > |RDT-1B|0.0308|0.0769|0.1385|0.1238|0.2154|0.2154|0.1335|
> > |**CogVLA**|**0.40(+0.32)**|**0.68(+0.60)**|**0.62(+0.48)**|**0.36(+0.24)**|**0.78(+0.56)**|**0.61(+0.39)**|**0.61(+0.48)**|
> >
> > Note: **For instance, the instruction "A burger on the table, but it feels like something's missing" from the "Select Drink" task implicitly requires the robot to fetch a Cola.**
> >
> > ------
> >
> > **(2.2) Quantitative Evaluation on Reasoning-Intensive Tasks**
> >
> > The LIBERO-Long benchmark emphasizes reasoning with extended temporal dependencies, which are central to human cognition. We evaluate CogVLA against over 50 state-of-the-art models (top 10 shown below; full results in the revised submission). CogVLA outperforms all existing methods.
> >
> > |Model|CogVLA|OpenVLA-OFT|UniVLA|PD-VLA|VOTE|UVA|LBP|Seer|OpenVLA-FreqPolicy|BEAST-F|
> > |-|-|-|-|-|-|-|-|-|-|-|
> > |SR(%)|95.4|94.5|92.0|91.7|91.0|90.0|88.6|87.7|87.6|86.4|
> >
> > ------
> >
> > **(2.3) Qualitative Analysis of Human-Like Visual Alignment**
> >
> > - In **Appendix D.2**, we visualize **attention heatmaps** produced by CogVLA to assess whether the perception module focuses on **instruction-relevant regions**.
> > - Additional visualizations from CALVIN, VLABench, and real-world settings will be included in the revised submission.
> >
> > ------
> >
> > **(2.4) Qualitative Analysis of Human-Like Error Correction Behavior**
> >
> > - In both simulation and real-world deployment, **CogVLA consistently retries failed actions**, despite no explicit supervision. In contrast, other models proceed without correction. We attribute this emergent behavior to CogVLA’s **structured intent modeling and observation-action grounding**, enabling reflective reassessment of action correctness.
> > - Such **self-initiated error detection and rollback** is rare in models without cognitively structured planning. **We will release video demonstrations on our project website.**
> >
> > ------
> >
> > **(2.5) Ablation of EFA-Routing → LFP-Routing → CAtten**
> >
> > **Table 4 of the paper** presents ablations validating the effectiveness of each module in our cognitively inspired architecture.
>
> ------
>
> ------
>
> ### W2: Limited comparative experiments.
>
> ### R2: We address this with a broader range of evaluations.
>
> **(1) Broader Benchmarking**
>
> > **(1.1) CALVIN Benchmark (ABC→D) Evaluation.**
> >
> > CogVLA demonstrates competitive performance. The following results are adapted from the UniVLA paper (**RSS 2025, May 2025**, Learning to Act Anywhere with Task-centric Latent Actions):
> >
> > |Method|1|2|3|4|5|Avg.Len.|
> > |-|-|-|-|-|-|-|
> > |RT-1|53.3|22.2|9.4|3.8|1.3|0.90|
> > |RoboFlamingo|82.4|61.9|46.6|33.1|23.5|2.48|
> > |SuSIE|87.0|69.0|49.0|38.0|26.0|2.69|
> > |GR-1|85.4|71.2|59.6|49.7|40.1|3.06|
> > |OpenVLA|91.3|77.8|62.0|52.1|43.5|3.27|
> > |CLOVER|96.0|83.5|70.8|57.5|45.4|3.53|
> > |RoboDual|94.4|82.7|72.1|62.4|54.4|3.66|
> > |UniVLA|95.5|85.8|75.4|66.9|**56.5**|3.80|
> > |**CogVLA**|**97.2**|**86.8**|**76.0**|**67.2**|56.2|**3.83**|
> >
> > Note: While CALVIN favors large-scale pretraining, CogVLA achieves state-of-the-art results by fine-tuning on ABC and testing on D **without any pretraining**.
> >
> > ------
> >
> > **(1.2) VLABench Evaluation.**
> >
> > Please refer to **(R1-2.1)**.
> >
> > ------
> >
> > **(1.3) Real-world Evaluation.**
> >
> > We further extended real-world experiments on the ALOHA platform **from 10 to 25 trials**, demonstrating the stability and consistency of CogVLA's performance.
> >
> > ||Cube→Plate+Toy→Bowl|Open+Place+Close|Step1+Step2+Step3|Avg SR|
> > |-|-|-|-|-|
> > |CogVLA|23/25+22/25|21/25+19/25+17/25|23/25+21/25+17/25|**74.7%**|
> >
>
> **(2) Consistency Analysis of Fluctuant**
>
> > **(2.1) Consistency Across Seeds**
> >
> > **Table 1 in Appendix shows CogVLA consistently low variance across LIBERO task subsets**, with an average success rate of 97.4% (mean ± std over 3 seeds), indicating strong generalization and reliability.
> >
> > |Method|Spatial|Object|Goal|Long|Average|
> > |-|-|-|-|-|-|
> > |CogVLA|98.5±0.5|98.8±0.4|96.5±0.6|95.2±1.1|**97.4±0.4**|
> >
> > ------
> >
> > **(2.2) Cascaded Stability**
> >
> > The CAtten module introduces a coupled attention mechanism that preserves fine-grained cross-modal alignment and coherent action planning, even with heavily pruned visual input—a capability beyond standard attention. **Its role is essential within the EFA-Routing → LFP-Routing → CAtten cascade and cannot be replaced by earlier stages.**
>
> ------
>
> ------
>
> ### W3: Lack of hardware variation in efficiency results.
>
> ### R3: Two main points as follows.
>
> > **(1) Efficiency Across Multiple GPU Configurations**
> >
> > As shown below, CogVLA achieves consistently superior performance across hardware variants:
> >
> > |Method|GPU|Train Cost↓|Infer Time↓|Throughput↑|
> > |-|-|-|-|-|
> > |OpenVLA-OFT|L40s|10.8h/10k|0.127s|63.0Hz|
> > |**CogVLA**|**L40s**|**7.2h/10k**|**0.088s**|**90.9Hz**|
> > |OpenVLA-OFT|A800|12.5h/10k|0.132s|60.6Hz|
> > |**CogVLA**|**A800**|**4.7h/10k**|**0.091s**|**87.9Hz**|
> >
> > Note: L40s results use reduced batch size. CogVLA benefits more from larger batches due to sparsity.
> >
> > ------
> >
> > **(2) Comparison with Other Acceleration Methods (LIBERO-Spatial)**
> >
> > **We report comparisons with two other acceleration methods in Table 6 of the paper.** Here, we further include MoD, where CogVLA demonstrates a clear advantage.
> >
> > |Method|CogVLA|FastV|SliME|MoD|
> > |-|-|-|-|-|
> > |SR(%)|98.6|88.2|77.6|81.0|
> >
>
> ------
>
> ------
>
> ### Minor W1: Generalization remains underexplored
>
> > **R1:** We added results on out-of-distribution instructions and unseen categories. See **W2-(1)** for the ABC→D evaluation on the CALVIN benchmark, where CogVLA shows strong zero-shot generalization.
>
> ### Minor W2: Missing explanations in figures
>
> > **R2:** Thank you for pointing this out. We’ve added missing annotations, revised Figure 1 and Equation (1), and improved overall clarity. A full revision will be submitted.
>
> ### Minor W3: Typographical error
>
> > **R3:** The typo has been corrected and all tables rechecked. We appreciate your careful feedback.
>
> ------
>
> ------
>
> We sincerely thank you for the insightful suggestions, which have been invaluable in refining the paper’s structure and empirical support. We hope our responses sufficiently address your concerns.

---

> > ### Author Response · Authors · 2025-08-04
> > **Invitation for Further Discussion**
> >
> > Dear Reviewer Bu5w,
> >
> > Thank you again for your valuable comments. We hope that our clarifications and additional experiments in the rebuttal have addressed each of your concerns. Should any questions remain unclear, we would appreciate the opportunity for further discussion.
> >
> > Best regards, Authors

---

> > > ### Comment · Area_Chair_y8tF · 2025-08-05
> > >
> > > Dear Reviewer  Bu5w,
> > >
> > > This is a gentle reminder to participate in the author discussion for your assigned paper(s). Engaging with authors is required before submitting the Mandatory Acknowledgement.
> > >
> > > The discussion deadline is August 8, 11:59 PM AoE. Please ensure you post at least one response in the discussion thread.
> > >
> > > Let me know if you encounter any issues.
> > >
> > > Best,
> > > Area Chair, NeurIPS 2025

---

> ### Author Response · Authors · 2025-08-08
>
> Dear Reviewer Bu5w,
>
> Thank you very much for taking the time and effort to respond to our rebuttal.
> However, we noticed that the discussion window in which you replied belongs to Reviewer ER8U.
> We are concerned that you may have mistakenly assumed the score of 5 given by Reviewer ER8U was your own initial score, and thus maintained your score under the impression that your concerns had already been well addressed. In fact, your original score was 3, and you mentioned in the "Questions" section that you would consider raising it if the first major weakness were thoughtfully addressed.
>
> We would be sincerely grateful if you could consider updating your score from 3 to 5 to reflect your positive feedback and appreciation of our paper and rebuttal.
>
> Best regards, Authors

---

> ### Comment · Reviewer_Bu5w · 2025-08-08
>
> Thanks for your response and related contents have well adressed my concern. I would like to raise my score.

---

> > ### Author Response · Authors · 2025-08-08
> >
> > Dear Reviewer Bu5w,
> >
> > Thank you very much for your kind response and for considering raising your score. We truly appreciate your review and positive feedback, which will help us improve the paper further.
> >
> > Best regards, Authors

---

> > ### Author Response · Authors · 2025-08-08
> >
> > Dear Reviewer Bu5w,
> >
> > We look forward to further discussion with you, and would also like to kindly remind you to submit the "Acknowledgement" and "Final rating" before the upcoming rebuttal deadline. Thank you again for your positive feedback.
> >
> > Best regards, Authors

---

### Official Review · Reviewer_iG5e · 2025-07-03

**Clarity:** 3
**Significance:** 2
**Originality:** 2
**Rating:** 4
**Confidence:** 3

**Summary:**

This paper introduces CogVLA, a vision-language-action (VLA) model that uses sparsification to improve efficiency and performance. They propose an Encoder-FiLM based aggregation rounting (EFA-Routing), a LLM-FiLM based pruning (LFP-Routing) and a VLA coupled attention (CAtten) to achieve this. They conduct experiments on LIBERO benchmark and real-world tasks, showing that CogVLA outperforms existing VLA models in terms of efficiency and performance.

**Questions:**

Answer all the questions in Weaknesses.

**Ethical Concerns:**

["NO or VERY MINOR ethics concerns only"]

**Final Justification:**

The rebuttal solves my concern about its improvement over the state-of-the-art VLA models on both efficiency and performance, so I raise my rating to 4.

**Limitations:**

yes

**Quality:**

3

**Strengths And Weaknesses:**

Strengths
1. The paper is well-written and structured, making it easy to follow the proposed approach.
2. The authors provide a clear motivation for the use of sparsification in VLA models.

Weaknesses
1. The methodology novelty is limited.
1) The CAtten seems already proposed in $\pi_0$. $\pi_0$ proposes to use bi-directional attention within each modality, i.e., the action decoding is actually parellel and requires only one feed-forward pass. The authors should clarify how their CAtten differs from $\pi_0$.
2) The proposed EFA-Routing and LFP-Routing focus on the same sparsification problem as in VLM models, which has been extensively studied in the literature. VLMs also study fine-grained instruction-driven reasoning, so the challenges faced by both VLA and VLM models are similar. The authors should clarify how their approach differs from existing methods in VLMs and why it is suitable for VLA models. There seems to be no specilized design for VLA models. The authors should clarify how their approach differs from existing methods in VLMs and why it is suitable for VLA models.
2. The experiments are not convincing.
1) The authors choose OpenVLA-style as the main baseline for efficiency evaluation, which is not a strong baseline. $\pi_0$ removes auto-regressive decoding and uses flow-matching to predict continuous action, which is effective and much more efficient than OpenVLA-style (70ms vs 250ms). The authors should include $\pi_0$ as a baseline to demonstrate the efficiency and effectiveness of their method.
2) The evaluation is conducted on small number of trials (10 for each task in the real world), which may not be sufficient to draw reliable conclusions. On LIBERO benchmark, the proposed method achieves 0.3% improvement over OpenVLA-OFT, which raises questions about whether this is due to randomness or the effectiveness of the proposed method. The authors should report the standard deviation of the results and conduct more trials to ensure the reliability of the results.

---

> ### Author Rebuttal · Authors · 2025-07-30
>
> We sincerely thank your constructive feedback. We appreciate the recognition of our writing clarity and the motivation behind introducing sparsification in VLA models. Below, we address each concern in detail.
>
> ###  W 1.1: The CAtten seems ... from $\pi_0$.
>
> ### R 1.1:  Two key distinctions are outlined below.
>
> > **(1) Attention Design and Semantic Consistency**
> >
> > Please refer to **Figure 3 (CAtten vs. other methods)** in the paper, which illustrates two key features of CAtten: i) a hierarchically structured attention mask, and ii) the injection of action intent from instruction into observation tokens via LLM-FiLM modulation. Below, we further provide a comparison with \$\pi\_0\$, and will include its attention visualization in the **updated Figure 3**.
> >
> > - $\pi_0$ adopts **bidirectional attention** within three independent blocks: `[Observation, Instruction]`, `[Robot State]`, and `[Action]`. In contrast, **CAtten employs unidirectional causal attention** within `[Observation, Instruction]` and `[Robot State]`, maintaining compatibility with pretrained LLMs and supporting temporally coherent reasoning.
> > - $\pi_0$ treats `[Observation, Instruction]` as a single attention block with bidirectional attention, which disrupts the directional flow of semantic reasoning. In contrast, **CAtten separates `[Observation]` and `[Instruction]` and injects action intent from `[Instruction]` into `[Observation]` via LLM-FiLM**, thereby avoiding this issue and enhancing cross-modal consistency.
> >
> > ------
> >
> > **(2) Action Decoding Architecture**
> >
> > - $\pi_0$ adopts a dual-system Mixture-of-Experts (MoE) design: a shared VLM backbone and an independent action expert. While action decoding leverages flow matching, it demands **1 forward pass through the VLM** and **10 through the action expert**.
> > - In contrast, CogVLA adopts a unified architecture where `[Observation, Instruction], [Robot State] and [Action]`are processed within **a single LLM context window, enhancing reasoning capability**. Actions are decoded as continuous outputs via L1 regression over action chunks, requiring only **a single forward pass**.
> > - Notably, $\pi_0$’s action expert operates outside the VLM context, constraining its generalization to diverse or long-horizon tasks. This limitation is reflected on the LIBERO-Long benchmark, where CogVLA delivers superior performance:
> >
> >     |             | $\pi_0$          | $\pi_0$+FAST     | CogVLA   |
> >     | ----------- | ---------------- | ---------------- | -------- |
> >     | LIBERO-Long | 85.2 **(−10.2)** | 60.2 **(−35.4)** | **95.4** |
>
> ------
>
> ------
>
> ### W 1.2: The proposed EFA-Routing and LFP-Routing ... VLA models.
>
> ### R 1.2: Three key innovations distinguish our design from prior sparsification methods in VLMs.
>
> > ||VLMs|CogVLA|Suitability for VLA|
> > |--|-|-|-|
> > |Decoding Paradigm|Autoregressive|Parallel decoding|**Reduces inference latency for real-time robot control**|
> > |Input-side Sparsification|Focus on instruction-relevant regions|**EFA-Routing**: Instruction-driven (1) Local anchoring; (2) Global action cues|**Aligns token selection with manipulation semantics**|
> > |In-LLM Sparsification|Tokens unaware of instructions; text-supervised pruning|**LFP-Routing**: (1) Intent via LLM-FiLM; (2) Action-pruned|**Tri-modal alignment for instruction-conditioned action**|
> > |Progressive Sparsification|Absent|**EFA-Routing→LFP-Routing** (bio-inspired)|**Reflects cognitive stages in manipulation (VAS→SMA)**|
> >
> > We elaborate on the design contributions below:
> >
> > **(1) Input-side Task-guided Sparsification (EFA-Routing)**:
> >
> > EFA-Routing explicitly establishes a strong Observation-Instruction-Action linkage in VLA tasks by extracting:
> >
> > - **Local object anchoring** via DINOv2 path;
> > - **Global action cues** via SigLIP path;
> > - These complementary features are adaptively fused through an Aggregation Router, which **balances their contributions**.
> >
> > Visualizations of heatmaps (**Appendix D.2**) demonstrate that the retained tokens are both manipulation task-relevant and spatially discriminative.
> >
> > ------
> >
> > **(2) In-LLM Instruction-driven Sparsification (LFP-Routing)**:
> >
> > LFP-Routing performs structured token pruning within the LLM, conditioned on instruction semantics and action supervision. Through **LLM-FiLM modulation**, it injects goal intent into the shared context, forming a unified **instruction-observation-action tri-modal** reasoning mechanism tailored for VLA, which is absent in conventional VLM approaches.
> >
> > ------
> >
> > **(3) Progressive and Biologically Motivated Sparsification**:
> >
> > - EFA-Routing and LFP-Routing operate in a progressive fashion (**EFA-Routing → LFP-Routing**), reflecting cognitive-stage alignment (**VAS → SMA**).
> > - This cognitive alignment enhances efficiency without compromising action consistency, outperforming other efficiency-oriented VLA models such as DeeR-VLA, MoLe-VLA, and RoboMamba on both computational and functional metrics.
> > - CogVLA demonstrates a more **human action-like design** philosophy aligned with the intrinsic characteristics of robotic manipulation.
> >
>
> ------
>
> ------
>
> ### W 2.1: The authors choose OpenVLA-style ... their method.
>
> ### R 2.1: We position CogVLA as a cognitively aligned evolution of single-system VLA architectures.
>
> > In end-to-end VLA models, OpenVLA-style and $\pi_0$-style architectures represent two dominant paradigms, where OpenVLA excels in reasoning and $\pi_0$ offers higher inference efficiency.
> >
> > **(1) We include $\pi_0$ as a baseline for efficiency comparison as shown below.**
> >
> > |Method|LLM|Train Steps↓|Training Time (per 10k)↓|Relative Training Cost↓|Latency↓|
> > |-|-|-|-|-|-|
> > |$\pi_0$|2.6B|800–1200k|4.3 h|~14.1×|**0.078 s**|
> > |$\pi_0$+FAST|2.6B|80–200k|6.3 h|~2.9×|0.329 s|
> > |OpenVLA|7B|-|-|-|0.254 s|
> > |OpenVLA-OFT|7B|50–150k|12.5 h|~4.0×|0.132 s|
> > |**CogVLA**|**7B**|**50–80k**|**4.7 h**|**1 (baseline)**|**0.091 s**|
> >
> > - For **training efficiency**, compared to OpenVLA-OFT and CogVLA, $\pi_0$ requires an order of magnitude more steps.
> > - For **inference efficiency**, while CogVLA is slightly slower than $\pi_0$, it uses a larger 7B backbone (vs. $\pi_0$’s 2.6B), yet still achieves the best overall trade-off between performance and efficiency.
> > - As $\pi_{0.5}$ has not been open-sourced, we do not include efficiency comparisons with it.
> >
> > **(2) We include $\pi_0$ and $\pi_{0.5}$ as baselines for performance comparison on LIBERO benchmark.** Despite partial decoupling or gradient blocking, these variants still suffer from weakened semantic consistency and reduced reasoning capacity, especially on long-horizon tasks (LIBERO-Long):
> >
> > |Method|LIBERO-Spatial|LIBERO-Object|LIBERO-Goal|LIBERO-Long|LIBERO-Overall|
> > |-|-|-|-|--|-|
> > |$\pi_0$|96.8|**98.8**|95.8|85.2 **(−10.2)**|94.2|
> > |$\pi_0$+FAST|96.4|96.8|88.6|60.2 **(−35.4)**|85.5|
> > |$\pi_{0.5}$(scratch)|96.6|97.2|94.6|84.8 **(−10.4)**|92.7|
> > |$\pi_{0.5}$(gen. model)|98.0|97.8|95.6|85.8 **(−9.4)**|96.0|
> > |**CogVLA**|**98.6**|**98.8**|**96.6**|**95.4**|**97.4**|
> >
>
> ------
>
> ------
>
> ### W 2.2: The evaluation is conducted ... the results.
>
> ### R 2.2: We provide additional experimental validation to address this concern.
>
> > **(1) Extended Real-World Trials**
> >
> > - We have expanded the real-world evaluations **from 10 to 25 trials** under the same training setup. Results are shown below:
> >
> >     |Method|Cube→Plate+Toy→Bowl|Open+Place+Close|Step1+Step2+Step3|Avg SR|
> >     |--|--|-|--|-|
> >     |OpenVLA-OFT|19/25+17/25|20/25+16/25+13/25|18/25+15/25+11/25|54.7%|
> >     |CogVLA|23/25+22/25|21/25+19/25+17/25|23/25+21/25+17/25|**74.7%**|
> >
> > ------
> >
> > **(2) Statistical Reliability**
> >
> > - We report standard deviation across three seeds on the LIBERO benchmark in **Appendix C.1** to ensure statistical reliability:
> >
> >     ||Spatial|Object|Goal|Long|Average|
> >     |--|--|-|--|-|-|
> >     |**CogVLA**|98.5±0.5|98.8±0.4|96.5±0.6|95.2±1.1|**97.4±0.4**|
> >
> >     The low standard deviation (±0.4) confirms that CogVLA's performance advantage is consistent and not attributable to random variation.
> >
> > ------
> >
> > **(3) Broader Benchmarking on VLABench**
> >
> > - VLABench is a recently benchmark designed to test **world knowledge and common-sense reasoning**. We compare CogVLA with the official baselines reported in the VLABench paper. Across all subtasks, CogVLA delivers dramatic improvements over baselines.
> >
> >    |Method|Select Book|Select Drink|Select Fruit|Select Tube|Add Condiment|Insert Flower|Overall|
> >    |-|--|-|-|-|-|-|-|
> >     |Octo|0.0000|0.0000|0.0000|0.0154|0.0154|0.0000|0.0051|
> >     |OpenVLA|0.0769|0.0846|0.0462|0.0769|0.1238|0.1385|0.0911|
> >     |RDT-1B|0.0308|0.0769|0.1385|0.1238|0.2154|0.2154|0.1335|
> >     |**CogVLA**|**0.40(+0.32)**|**0.68(+0.60)**|**0.62(+0.48)**|**0.36(+0.24)**|**0.78(+0.56)**|**0.61(+0.39)**|**0.61(+0.48)**|
>
> We hope these clarifications adequately address your concerns. And we remain open to further suggestions and will actively incorporate them in future revisions.

---

> > ### Author Response · Authors · 2025-08-04
> > **Invitation for Further Discussion**
> >
> > Dear Reviewer iG5e,
> >
> > Thank you again for your valuable comments. We hope that our clarifications and additional experiments in the rebuttal have addressed each of your concerns. Should any questions remain unclear, we would appreciate the opportunity for further discussion.
> >
> > Best regards, Authors

---

> > > ### Comment · Area_Chair_y8tF · 2025-08-05
> > >
> > > Dear Reviewer iG5e,
> > >
> > > This is a gentle reminder to participate in the author discussion for your assigned paper(s). Engaging with authors is required before submitting the Mandatory Acknowledgement.
> > >
> > > The discussion deadline is August 8, 11:59 PM AoE. Please ensure you post at least one response in the discussion thread.
> > >
> > > Let me know if you encounter any issues.
> > >
> > > Best, Area Chair, NeurIPS 2025

---

### Official Review · Reviewer_ER8U · 2025-07-03

**Clarity:** 3
**Significance:** 3
**Originality:** 3
**Rating:** 5
**Confidence:** 3

**Summary:**

The authors present CogVLA, a novel Vision-Language-Action framework designed to address a critical bottleneck in the field: the immense computational cost and inefficiency of current state-of-the-art models. CogVLA introduces a cognition-aligned Vision-Language-Action framework that mirrors the human perceptual–reasoning–control loop through a three-stage, instruction-driven design. Stage 1 (Encoder-FiLM Aggregation Routing) embeds task instructions into the vision encoder to aggregate and compress visual tokens; Stage 2 (LLM-FiLM Pruning Routing) injects action intent into the language model to prune instruction-irrelevant tokens; and Stage 3 (V-L-A Coupled Attention) fuses the compressed multimodal context with bidirectional decoding for coherent, parallel action planning. By progressively pruning vision and language tokens and then fusing the compressed multimodal context, it cuts vision tokens 8×, FLOPs 3.1×, and training cost 2.5×, yet still achieves state-of-the-art success rates on LIBERO (97.4 %) and real-world ALOHA tasks (70 %). The work shows that instruction-aware sparsification can boost both efficiency and performance, advancing scalable embodied AI.

**Questions:**

None.

**Ethical Concerns:**

["NO or VERY MINOR ethics concerns only"]

**Final Justification:**

Thanks to the authors for their detailed response to my concerns. The additional experiments provided effectively address the concerns I raised. As my previous rating 5 already adequately reflects the quality of the paper, I will keep my rating.

**Limitations:**

None.

**Paper Formatting Concerns:**

None.

**Quality:**

3

**Strengths And Weaknesses:**

Pros:
1. The paper tackles one of the most pressing issues in embodied AI—the trade-off between model capability and computational feasibility. The proposed solution offers a path toward deploying powerful VLA models on resource-constrained platforms.
2. The paper delivers on its promises. Achieving state-of-the-art performance while simultaneously delivering order-of-magnitude improvements in efficiency is a rare and significant contribution.

Cons:
1. The model is tested on structured tasks from LIBERO and ALOHA. It is unclear how well the instruction-driven sparsification would generalize to tasks with fundamentally different structures or those requiring more implicit, common-sense reasoning not explicitly cued by the instruction.
2. Following #1, the entire framework is predicated on the instruction being an accurate and sufficient guide. The system's performance would likely degrade sharply with ambiguous, or incomplete instructions. An analysis of this failure mode would strengthen the paper.
3. The model introduces new, important hyperparameters, particularly the token retention ratio β in LFP-Routing and the sparsification ratio split between Stage 1 and 2 (Table 5). The paper demonstrates a good choice but does not provide a deeper analysis of the model's sensitivity to these parameters.

---

> ### Author Rebuttal · Authors · 2025-07-30
>
> We thank your thoughtful feedback and for acknowledging our contribution toward balancing computational efficiency and task performance. Below, we address each concern in detail:
>
> ### Cons 1: The model is tested ... instruction.
>
> ### R1: We address this concern from two perspectives.
>
> > **(1) Generalization on structurally distinct tasks**:
> >
> > To evaluate CogVLA's generalization under underspecified and implicit instruction settings, we conducted additional experiments on **VLABench**, a recently introduced open-source benchmark for language-conditioned manipulation. Unlike prior benchmarks, VLABench features:
> >
> > - tasks requiring commonsense and world knowledge transfer,
> > - **natural, non-templated instructions with implicit human intent**, and
> > - joint evaluation of action policy and language understanding.
> >
> > For example, in the 'Select Book' suite, one task is: “**Take 'Pride and Prejudice' from the bookshelf.**” As shown below, CogVLA outperforms all baselines by a large margin:
> >
> > |Method|Select Book|Select Drink|Select Fruit|Select Tube|Add Condiment|Insert Flower|Overall|
> > |------|-----------|------------|------------|-----------|--------------|--------------|--------|
> > |Octo|0.0000|0.0000|0.0000|0.0154|0.0154|0.0000|0.0051|
> > |OpenVLA|0.0769|0.0846|0.0462|0.0769|0.1238|0.1385|0.0911|
> > |RDT-1B|0.0308|0.0769|0.1385|0.1238|0.2154|0.2154|0.1335|
> > |**CogVLA**|**0.40(+0.32)**|**0.68(+0.60)**|**0.62(+0.48)**|**0.36(+0.24)**|**0.78(+0.56)**|**0.61(+0.39)**|**0.61(+0.48)**|
> >
> > ------
> >
> > **(2) Real-world tests under abstract instructions**:
> >
> > In addition to the “T-shirt Folding” task (Table 2 of the main paper), which requires commonsense reasoning to interpret an abstract goal, we introduce two further ALOHA experiments involving implicit and underspecified instructions:
> >
> > - **“Clean up the table.”**: No objects or targets specified; requires collecting all clutter.
> > - **“Take the toy out of the drawer.”**: The toy is initially invisible; no explicit drawer-related actions (e.g., “open”) are mentioned.
> >
> > The results are below:
> >
> > ||Step 1|+Step 2|+Step 3|
> > |-|-------|--------|--------|
> > |Fold the T-shirt.|21/25|20/25|17/25|
> >
> > ||All Cleaned|1 Left↓|2 Left↓|≥3 Left↓|
> > |-|-----------|------|------|--------|
> > |Clean up the table.|20/25|2/25|1/25|2/25|
> >
> > | |Open Drawer|+Take Out Toy|
> > |-|-----------|-------------|
> > |Take the toy out of the drawer.|18/25|15/25|
> >
> > Note: The number of trials has been increased from 10 to 25 for stability. We will include visualizations of these new tasks in the revised submission.
> >
>
> ------
>
> ------
>
> ### Cons 2: Following #1 ... paper.
>
> ### R2: Analyzing robustness under ambiguous or incomplete instructions.
>
> > We conducted a failure analysis using the **VLABench** benchmark, which naturally features instructions with **implicit, metaphorical, or underspecified intent**. Below we present representative failure cases from three task suites, followed by successful examples where **CogVLA outperforms OpenVLA**, illustrating its improved reasoning capacity.
> >
> > **Failure Case Analysis of CogVLA:**
> >
> > 1. （Select Drink）instruction：“I'm craving some adrenaline for my upcoming gaming marathon.”
> >
> > 2. （Select Fruit ）instruction：“I want a fruit that symbolizes knowledge and education.”
> >
> > 3. （Select Fruit ）instruction：“The chicken wings are almost done; they just need a final touch to perfect their flavor profile.”
> >
> >     |ID|Ground Truth|Predicted|Failure Type|Analysis|
> >     |--|-------------|----------|-------------------------|-----------------------------------------------------------|
> >     |1|Monster|Juice|Metaphorical understanding|Metaphorical “adrenaline” poses a mild challenge; suggests improvement in **metaphor understanding and intent grounding**.|
> >     |2|Apple|Random|Symbolic reasoning|Symbolic “apple” shows gap in **analogy and cultural grounding**; room for enhancement.|
> >     |3|BBQ Sauce|Salt|Multi-target ambiguity|“Final touch” is vague; various condiments apply—**more context or user preference** would help disambiguate.|
> >
> > ------
> >
> > **Success Cases of CogVLA over OpenVLA:**
> >
> > 1. （Select Drink）instruction：“I want a refreshing beverage with natural fruit sugars. ”
> > 2. （Select Fruit ）instruction：“I need a fruit that is typically yellow when ripe.”
> > 3. （Add Condiment）instruction：“To enrich the flavors in this dish, we need something with a tomato base that's slightly sweet yet vinegary. Please add it.”
> >
> >     |ID|CogVLA Pred.|OpenVLA Pred.|Reasoning Type|Analysis|
> >     |--|--------------|----------------|------------------------|-----------------------------------------------------------|
> >     |1|Juice|Cola|Functional semantics|**CAtten enhances grounding; “fruit sugars” better aligns with juice than soda.**|
> >     |2|Banana|Pear|Commonsense reasoning|**EFA-Routing preserves yellow-related tokens, guiding correct banana grounding.**|
> >     |3|Ketchup|Salt|Multi-attribute reasoning|**CogVLA integrates cues (“tomato,” “sweet,” “vinegary”) → ketchup; shows strong compositional reasoning.**|
> >
> >     Note: The OpenVLA baseline is from the official VLABench repository.
> >
> > We will include frame-wise heatmaps and pruning-confidence curves for the above cases in the revised appendix, along with additional examples across task suites. These failure modes also reveal future research directions: improving metaphor and event-grounded reasoning, integrating symbolic knowledge, and modeling user preference under ambiguous intent.
> >
>
> ------
>
> ------
>
> ### Cons 3: The model introduces ... parameters.
>
> ### R3: Clarifications on hyperparameter design are as follows.
>
> > **(1) Hyperparameter β in LFP-Routing**:
> >
> > We use a shifted cosine schedule (details in **Appendix A.1**) to modulate token retention across layers, with β constrained to [0.05, 0.85] to avoid degenerate pruning. We conducted two experiments varying β, and the results exhibit high stability:
> >
> > |Clamp Range|Approx.Pruning Rate|Spatial SR|
> > |-----------|-------------------|-----------|
> > |[0.05,0.85]|0.533|98.6|
> > |[0.10,0.80]|0.536|98.6|
> > |[0.15,0.75]|0.539|98.4|
> >
> > ------
> >
> > **(2) Sparsification ratio between Stage 1 and Stage 2**:
> >
> > To achieve a target 8× sparsification (12.5% retention), we fix Stage 2 at 0.533 (in line with the β experiments above) and vary Stage 1 rates. The results show consistent performance:
> >
> > |Stage1 Aggregation|Stage2 Pruning|Spatial SR|
> > |------------------|---------------|-----------|
> > |0.250|0.533|98.6|
> > |0.234|0.533|98.4|
> > |0.266|0.533|98.6|
> >
> > Note: Total retention ≈ 0.125. Stage 2 fixed; Stage 1 varied.
> >
> > ------
> >
> > **(3) Ablation Study of Hyperparameters in Appendix C.3**:
> >
> > These results validate that the chosen hyperparameters enable substantial sparsity while preserving task performance.
> >
> > |Stage1|Stage2|Total Sparsification|Spatial SR|
> > |------|------|--------------------|-----------|
> > |2×|2×|4×|96.2|
> > |4×|4×|16×|93.2|
> > |2×|4×|8×|94.6|
> > |**4×**|**2×**|**8×**|**98.6**|
> >
>
>
> We sincerely thank you again for the constructive feedback, which has significantly contributed to improving our work. We remain open to further suggestions and will actively incorporate them in future revisions.

---

> > ### Author Response · Authors · 2025-08-04
> > **Invitation for Further Discussion**
> >
> > Dear Reviewer ER8U,
> >
> > Thank you again for your valuable comments. We hope that our clarifications and additional experiments in the rebuttal have addressed each of your concerns. Should any questions remain unclear, we would appreciate the opportunity for further discussion.
> >
> > Best regards, Authors

---

> > > ### Comment · Area_Chair_y8tF · 2025-08-05
> > >
> > > Dear Reviewer ER8U,
> > >
> > > This is a gentle reminder to participate in the author discussion for your assigned paper(s). Engaging with authors is required before submitting the Mandatory Acknowledgement.
> > >
> > > The discussion deadline is August 8, 11:59 PM AoE. Please ensure you post at least one response in the discussion thread.
> > >
> > > Let me know if you encounter any issues.
> > >
> > > Best,
> > > Area Chair, NeurIPS 2025

---

> ### Author Response · Authors · 2025-08-08
>
> Dear Reviewer Bu5w,
>
> Thank you very much for taking the time and effort to respond to our rebuttal. However, we noticed that this discussion window belongs to Reviewer ER8U. We are concerned that you may have mistakenly assumed the score of 5 given by Reviewer ER8U was your own initial score, and thus maintained your score under the impression that your concerns had already been well addressed. In fact, your original score was 3, and you mentioned in the "Questions" section that you would consider raising it if the first major weakness were thoughtfully addressed.
>
> We would be sincerely grateful if you could consider updating your score from 3 to 5 to reflect your positive feedback and appreciation of our paper and rebuttal.
>
> Best regards, Authors

---

### Comment · Area_Chair_y8tF · 2025-08-02

Dear Authors and Reviewers,

Thank you to the authors for the detailed rebuttal.

Reviewers, please read the responses carefully and post your reply as soon as possible to allow for meaningful discussion. Ideally, all reviewers should respond so the authors know their feedback has been considered.

Best regards,
AC

---

### Note · Authors · 2025-08-12

Dear Area Chairs,

We would like to extend our sincere gratitude for your dedication and effort in overseeing the review process. We are also grateful to all reviewers for taking the time and effort to review our work.

1. **Reviewer ER8U: Positive score of 5.** They noted that our method *“achieving state-of-the-art performance while simultaneously delivering order-of-magnitude improvements in efficiency is a rare and significant contribution.”* We further provided detailed responses and additional experimental evidence.

2. **Reviewer Bu5w: Positive score $\geq$ 4 .** They highlighted *“Significant gains in both efficiency and performance”* and *“Original method design,”* as well as the paper’s clear structure and writing. We addressed their concerns in detail, to which they responded that the *“contents have well addressed my concern.”*

3. **Reviewer EsjF: Positive score of 4.** They stated that our method is both effective and efficient, with paper’s clear writing. We provided detailed responses, and the reviewer maintained their positive score.

4. **Reviewer iG5e**: The current score from this reviewer is **not yet available** to us. Their concerns and our corresponding responses are summarized below:

    > | Concerns | Our Rebuttal|
    > | - | -|
    > | “The authors should clarify how their CAtten differs from π₀.” | We outlined multiple key differences in our rebuttal to clarify their fundamental distinction.|
    >| “The authors should clarify how their approach differs from existing methods in VLMs and why it is suitable for VLA models.” | We provided detailed reasoning showing that our module design and human action–inspired architecture are original and tailored specifically for VLA models. |
    > | “The authors should include π₀ as a baseline to demonstrate the efficiency and effectiveness of their method.” | We added π₀ as an additional baseline, where CogVLA outperforms π₀ in nearly all performance and efficiency metrics, with only a minor latency gap due to our larger 7B backbone compared to π₀’s 2.6B. We also included π₀+FAST and π₀.₅ as baselines, with CogVLA remaining the top performer. |
    >| “The authors should report the standard deviation of the results and conduct more trials to ensure reliability.” | We have included the requested additional real-world trials, and the standard deviations are reported in the appendix. |

We sincerely hope these points will be taken into account in the final decision.

Best regards, Authors

---

### Decision · Program_Chairs · 2025-09-17

**Decision:**

Accept (poster)

**Comment:**

This is a solid paper that introduces CogVLA, a cognition-aligned framework for vision-language-action tasks. The main strengths highlighted by reviewers are the novelty of the design, the strong efficiency–performance trade-off, and the clarity of the presentation.

The main concerns raised were about (i) how CAtten differs from π₀, (ii) how the approach compares to existing VLM-based methods, (iii) whether the baselines were sufficient, and (iv) the reliability of results. The authors provided an excellent rebuttal, clarifying conceptual differences, adding stronger baselines, and reporting additional trials with standard deviations. These additions directly addressed the reviewers’ questions and helped build confidence in the contribution.

After the rebuttal, all reviewers agreed the concerns were resolved and maintained positive scores (5, 4, 4, 4). Overall, the paper makes a novel and significant contribution, and given the strong rebuttal and consensus, I recommend acceptance.